# COMPLETED HYPERPARAMETER TRANSFER ACROSS MODULES, WIDTH, DEPTH, BATCH & DURATION

**Bruno Mlodozeniec**[*]
University of Cambridge

**Pierre Ablin**
Apple

**Louis Bethune**
Apple

**Dan Busbridge**
Apple

**Michal Klein**
Apple

**Jason Ramapuram**
Apple

**Marco Cuturi**
Apple

## ABSTRACT

Hyperparameter tuning can dramatically impact training stability and final performance of large-scale models. Recent works on neural network *parameterisations*, such as $\mu$P, have enabled transfer of optimal global hyperparameters across model sizes. These works propose an empirical practice of search for optimal *global* base hyperparameters at a small model size, and transfer to a large size. We extend these works in two key ways. To handle scaling along most important scaling axes, we propose the **Complete**[(d)] Parameterisation that unifies scaling in width & depth — using an adaptation of *CompleteP* (Dey et al., 2025) — as well as in *batch-size* and *training duration*. Secondly, with our parameterisation, we investigate per-module hyperparameter optimisation and transfer. We characterise the empirical challenges of navigating the high-dimensional hyperparameter landscape, and propose practical guidelines for tackling this optimisation problem. We demonstrate that, with the right parameterisation, hyperparameter transfer holds even in the per-module hyperparameter regime. Our study covers an extensive range of optimisation hyperparameters of modern models: learning rates, AdamW parameters, weight decay, initialisation scales, and residual block multipliers. Our experiments demonstrate significant training speed improvements in Large Language Models with the transferred per-module hyperparameters.

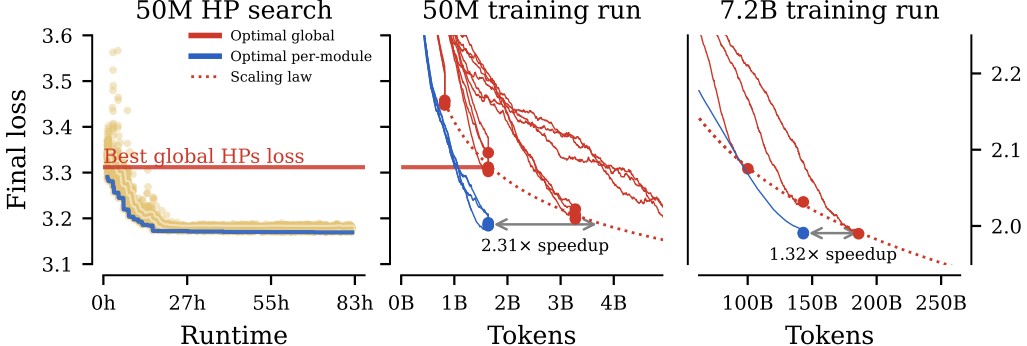

Figure 1: **(Left):** We optimise hyperparameters at a small 50M parameters/1.6B tokens scale (learning rate, initialisation scale, Adam $\varepsilon, \beta_1, \beta_2$ and weight decay) with an evolutionary strategy. These hyperparameters (HPs) can be either optimised *globally* (one value across the entire model), or *per-module* (13 module types, some tuned per depth). **(Middle):** At the 50M scale, the per-module HPs are $2.3\times$ faster to reach the same level of performance as the global HPs. **(Right):** Our new Complete[(d)] Parameterisation enables *direct* transfer (without *any* subsequent tuning) to a $\sim 10000\times$ larger FLOP budget. While we can transfer both optimal global and per-module HPs, the per-module HPs retain a $1.32\times$ speed-up and improved benchmark performance (see Figure 9) at 7.2B scale .

---

[*]Work done at Apple.

## 1 INTRODUCTION

The remarkable success of large transformer-based models (Vaswani et al., 2017) has been driven by scaling up model size and data (Kaplan et al., 2020). However, to get the most out of these large-scale training runs, or to even successfully complete them at all, a multitude of hyperparameters (HPs), such as learning rates, weight decay, or initialization scales, must be carefully set.

**Parameterisation.** To mitigate this HP tuning cost, recent works have introduced principled *parameterisations* — such as the $\mu$-parameterisation ($\mu$P) (Yang et al., 2022) — that enable the *transfer* of hyperparameters from smaller, cheaper-to-train models to their large-scale counterparts. Effectively, these parameterisations automatically adapt an HP for any tensor, depending on its type and the change in width. This process has been extended to handle changes in depth with Depth-$\mu$P (Yang et al., 2024) and further investigated for width and depth transfer in transformers with CompleteP (Dey et al., 2025). These methods have been demonstrated to successfully transfer optimal *global* hyperparameters like the learning rate. However, two questions remain open:

**Per-module HPs**: *Can tuning HPs on a per-module basis give significant gains, and do the per-module HPs empirically transfer with the right parameterisation?* Given the significant performance improvements from well-optimised global HPs, it is natural to consider optimising them on a finer-grained scale as well. When scaling up a model with $\mu$P, Depth-$\mu$P or CompleteP, different tensors will receive different HPs depending on their architectural role – for instance, the learning rates for the embedding layers have to be scaled differently from those for the hidden weights. It is therefore reasonable to expect that different modules, or tensors, could benefit from independent hyperparameter tuning. Put differently, there is little reason to believe the optimal per-module HPs should all collapse to the same value at *some* base width at which we optimise them.

**Scaling modalities**: *How should we parameterise training for scaling beyond just model size?* $\mu$P and extensions are only derived for transfer in model width & depth. However, model size is only *one* of the scaling axes utilised for training better models. Scaling data and batch-size (the latter often being necessary to facilitate training on a larger corpus) is also critical, but the HP transfer will not hold along these axes if relying solely on $\mu$P (Sections 2.2 and 2.3).

In this work, we systematically investigate the *transfer* of per-module hyperparameters across various scaling modalities. Per-module hyperparameters introduce a scaling challenge: tuning per-module HPs basis creates an explosion of the dimensionality of the search space, making it truly intractable at a large scale. We demonstrate that a practical methodology of optimising at the small-scale, and using an HP-transfer-enabling parameterisations works. We perform the expensive, high-dimensional search for optimal per-module HPs on a small proxy model, and demonstrate the transfer of the optimal HPs to a large target model. Our contributions are:

- **Complete$^{(d)}$P for width & depth transfer**. We refine the CompleteP parameterisation from Dey et al. (2025), extending it to modern Transformer components like Query-Key Normalisation (Henry et al., 2020). We further identify and rectify minor issues in the original formulation. We illustrate the resulting parameterisation permits robust hyperparameter transfer for all theoretically-motivated variants of depth scaling ($\alpha \in \left[\frac{1}{2}, 1\right]$).
- **Complete$^{(d)}$P for transfer in new scaling axes**. We systematically study transfer beyond model size, including in token horizon and batch size. We unify our adaptation to CompleteP with batch-size and token horizon scaling rules. For batch-size scaling, we adapt the SDE reparameterisation in (Malladi et al., 2022) to AdamW, extending the prescription to scaling of *weight-decay*. We further propose a new SDE-inspired scaling rule for transfer in the token horizon.
- **Per-module HP transfer.** We empirically demonstrate that *hyperparameter transfer holds for per-module hyperparameters* with the right parameterisation. Optimising per-module hyperparameters at a small scale yields significant training speed-ups that persist after transfer to a larger scale.
- **A practical recipe to find per-module HPs.** We empirically characterise the per-module hyperparameter optimisation landscape. We highlight its challenging nature, marked by sharp "cliffs" where training diverges. These characteristics make it highly inefficient to use random search and can prove challenging for default Bayesian Optimisation. We show that the landscape is close to "invex", and that *local* search strategies work well.

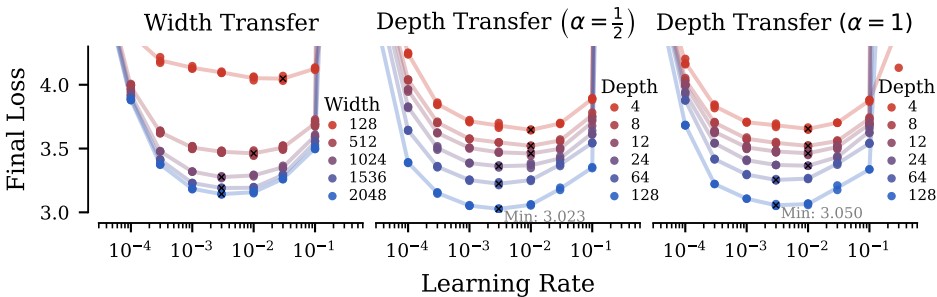

Figure 2: Hyperparameter transfer for global learning rate across depth and width. Each setting is run with three independent seeds.

## 2 HYPERPARAMETER TRANSFER

In this section, we describe the hyperparameter transfer modalities we consider, and the principles that we follow to adjust HPs while varying other aspects of the training configuration.[1] We first describe hyperparameter transfer in model size (width and depth) in Section 2.1, where we introduce a variant of the CompleteP parameterisation (Dey et al., 2025). In Section 2.2, we describe principles we follow for hyperparameter transfer across batch-size. Lastly, we consider hyperparameter transfer in the number of training tokens in Section 2.3, illustrating that optimal HPs do not transfer out of the box across token horizons. For an extended discussion of related work, see Appendix E.

**Experimental Setup**   All experiments are conducted using a decoder-only transformer model (Radford et al., 2019; Phuong & Hutter, 2022) on the RedPajama dataset (Weber et al., 2024). We use a modern transformer variant with pre-normalisation, Query-Key norms (Henry et al., 2020), trained with a mixture of cross-entropy and Z-loss (de Brébisson & Vincent, 2016). We always train with a cosine schedule. As a performance metric, we always report the final validation loss on the pre-training data, which is a strong indicator of downstream performance (Hoffmann et al., 2022; Chen et al., 2025). For remaining training and architecture details, see Appendix D.

### 2.1 HYPERPARAMETER TRANSFER ACROSS MODEL SIZE

The core idea underlying hyperparameter transfer across models of different sizes is to view finite-size models as discretisations of the infinite-size limit. Intuitively, two models of different sizes that are both sufficiently close to the same infinite limit will behave similarly. If they are sufficiently close over a set of considered hyperparameters, then they should share similar optimal hyperparameters.

The challenge is that, depending on the *parameterisation* — i.e. the rules for adjusting the hyperparameters as a function of size — we can obtain different infinite width or depth limits with fundamentally different behaviours (Yang & Hu, 2021). Most of these limits are pathological in various ways. For instance, the Standard Parameterisation (SP) (Sohl-Dickstein et al., 2020) leads to the features blowing up with size, whereas the Neural Tangent Parameterisation (Jacot et al., 2018) results in a lack of *feature learning* (Yang & Hu, 2021). $\mu$P was identified by Yang & Hu (2021) as the unique parameterisation for Stochastic Gradient Descent (and later for a broad class of adaptive algorithms (Yang & Littwin, 2023)) that precludes the emergence of such pathologies at scale.

In this work, we build upon CompleteP (Dey et al., 2025), which itself is an adaptation of Depth-$\mu$P (Yang et al., 2024) to transformers, to which we make several adaptations. These new scaling rules, which we call **Complete$^{(d)}$P**, are summarised in Table 1. Firstly, we extend the parameterisation to Query-Key (QK) normalisation layers (Henry et al., 2020), which have become a staple in modern transformer implementations (Yang et al., 2025; Dehghani et al., 2023). The challenge of QK norms is that, unlike any other component in transformers, these layers share weights across transformer heads. If scaling in width is performed by increasing the number of heads while keeping the head dimension fixed (as was done in (Dey et al., 2025; Yang & Hu, 2021)), then QK norms introduce

---

[1]We disambiguate between *hyperparameters* (the settings we want to find optimal values for), and the *training configuration* (the settings we change to facilitate scaling: data size, model size, batch-size).

Table 1: **Parameterisation Comparison** as a function of width ($m_N$), depth ($m_L$), batch size ($m_B$) and token count/data size ($m_D$) ratios. For Complete$^{(d)}$P , differences to CompleteP (Dey et al., 2025) for width & depth scaling are shown alongside in gray.

| | Parameterisation: | | $\mu$P (Table 3) | Complete$^{(d)}$P | |
|---|---|---|---|---|---|
| **Multipliers** | MHA Residual | | $\mathbf{x} + \mathtt{MHABlock}(\mathbf{x})$ | $\mathbf{x} + m_L^{-\alpha}\mathtt{MHABlock}(\mathbf{x})$ | |
| | MLP Residual | | $\mathbf{x} + \mathtt{MLPBlock}(\mathbf{x})$ | $\mathbf{x} + m_L^{-\alpha}\mathtt{MLPBlock}(\mathbf{x})$ | |
| | Unemb. Fwd | | *Unaugmented* | *Unaugmented* $\scriptstyle [\times(m_N^{-1})]$ | |
| **Init. Variances** | Input Emb. | | | | |
| | Hidden weights | | $\times m_N^{-1}$ | $\times m_N^{-1}$ | |
| | Hidden biases/norms | $\sigma_{\mathrm{b}}^2$ | | | |
| | Unemb. LN | | | | |
| | Unemb. Weights | | $\times m_N^{-2}$ | $\times m_N^{-2}$ $\scriptstyle [\times 1]$ | |
| **Learning Rates** | Input Emb. | | | | |
| | Hidden weights | | $\times m_N^{-1}$ | $\times m_N^{-1} \times m_L^{\alpha-1}$ | |
| | Hidden biases/norm | $\eta_{\mathrm{b}}$ | | $\times m_L^{\alpha-1}$ | $\times\sqrt{\frac{m_B}{m_D}}$ |
| | Unemb. LN | | | | |
| | Unemb. weights | | $\times m_N^{-1}$ | $\times m_N^{-1}$ $\scriptstyle [\times 1]$ | |
| **AdamW $\epsilon$** | Hidden weights/biases/norms | | $\times m_N^{-1}$ | $\times m_N^{-1} \times m_L^{-\alpha}$ | |
| | QK norms | $\epsilon_{\mathrm{b}}$ | NA | $\times m_L^{-\alpha}$ $\scriptstyle [NA]$ | $\times\sqrt{\frac{m_D}{m_B}}$ |
| | Input Emb. | | $\times m_N^{-1}$ | $\times m_N^{-1}$ $\scriptstyle [\times 1]$ | |
| | Output weights/biases/norms | | | | |
| **Weight decay** | Hidden weights | | $\times m_N$ | $\times m_N$ | |
| | Unemb. weights | $\lambda_{\mathrm{b}}$ | $\times m_N$ | $\times m_N$ | $\times\sqrt{\frac{m_B}{m_D}}$ |
| | Rest | | $\times 1$ | $\times 1$ | |
| *AdamW* $(1-\beta_1)$ | | | $(1-\beta_{1,\mathrm{b}})$ | | $\times\frac{m_B}{m_D}$ |
| *AdamW* $(1-\beta_2)$ | | | $(1-\beta_{2,\mathrm{b}})$ | | $\times\frac{m_B}{m_D}$ |
| | | *Training iterations* $\propto \frac{m_D}{m_B}$ | | | |

weight-sharing across the scaled dimensions. This necessitates different scaling considerations than for regular normalisation layer multipliers or biases. The adjustments for AdamW (Loshchilov & Hutter, 2017) are shown in Table 1, which we justify in Appendix B.

Secondly, we note that Dey et al. (2025) mistakenly derived the wrong scaling for the AdamW $\epsilon$ scaling for the input embedding. We justify our modification in the Appendix B. Although the resulting modification is minor, we found that the lack thereof was sufficient to break a thorough sweep of the coordinate checks described by Yang et al. (2022) in our implementation.

Lastly, we eliminate the explicit scalar multiplier on the output of the final linear projection by reparameterising its effect into the learning rate and initialisation scale. This enables easily incorporating memory-efficient algorithms like Cut Cross-Entropy (Wijmans et al., 2025), which avoid materialising the full logit matrix, drastically reducing GPU memory requirements for modern large vocabulary models.

In Figure 2, we verify the HP transfer with Complete$^{(d)}$P across width and depth. An important factor in Depth-$\mu$P is the depth-dependent re-scaling factor for the residual connection in transformers ($\mathbf{h}^\ell$ the output of layer $\ell$, $\mathcal{F}_\ell$ the function applied to it):

$$\mathbf{h}^{\ell+1} = \mathbf{h}^\ell + m_L^{-\alpha}\mathcal{F}_\ell(\mathbf{h}^\ell), \quad \ell \in \{1, \dots, L\}$$

which is governed by a single parameter $\alpha \in [0.5, 1]$. We make the following observation:

> Complete$^{(d)}$P with $\alpha = \frac{1}{2}$ and $\alpha = 1$ permits hyperparameter transfer across depth.

Our parameterisation seems to allow for HP transfer with all theoretically justified values of $\alpha \in \left[\frac{1}{2}, 1\right]$. This is in contrast to the findings of Dey et al. (2025) who notice a degradation of transfer for $\alpha = 0.5$. The added QK norms in our implementation improve stability (see Figure 13

for a comparison without); however, removing them does not lead to the breakdown of transfer reported by Dey et al. (2025). We note that in their publicly-released reference implementation, they apply the same AdamW's $\epsilon$ to all weights (including embeddings), against their own recommendation. Interestingly, the optimal loss is slightly better for the largest model for $\alpha = \frac{1}{2}$, potentially suggesting that the theoretical arguments for this parameterisation on the basis of *feature diversity* (Yang et al., 2024) might be beneficial in a language transformer context.

## 2.2 Hyperparameter transfer across batch-size

Model size and dataset size are two levers to achieve lower loss – increasing each predictably leads to model improvements, as implied by scaling laws (Kaplan et al., 2020; Hoffmann et al., 2022). To avoid the training time blowing out of proportions, one typically wants to make use of extensive parallelisation, such as that controlled by the batch-size. However, the set of practically usable batch-sizes is often heavily constrained by the memory footprint of a model of a given size, and the specific parallel compute architecture available for training. In practice, the availability of compute can often change from month to month or week to week, requiring adjustments. Furthermore, these requirements might be different when tuning hyperparameters vs. when scaling up. For smaller hyperparameter sweep runs, a smaller batch-size is often desirable to reduce the per-run memory footprint and enable running parallel HP search. When training a larger model, one typically wants to adjust the batch-size to make use of all available compute for a single run. Unfortunately, as with scaling model size or training duration, hyperparameters do not transfer across batch-sizes without further reparameterisation. In this work, we transfer hyperparameters across batch-size via a similar limiting argument as for transfer across model size. In particular, we follow the principles for batch-size transfer laid out in Malladi et al. (2022), and extend them to weight-decay in AdamW.

**Training as discretising an SDE**. The goal of the SDE reparameterisation approach is to parameterise hyperparameters such that training trajectories remain approximately invariant to batch-size changes when measured in "normalised time" (proportional to data observed). However, exact invariance is generally impossible. We therefore aim for invariance in the limit of infinitely small "effective" step sizes (step sizes in normalised time), where the discrete optimizer iterates (e.g., Adam or SGD) converge to a continuous stochastic process (often described by a Stochastic Differential Equation (SDE) (Malladi et al., 2022)). By identifying the parameterisation that ensures a consistent and sensible SDE limit, we can derive reparameterisation rules where the optimizer acts as a stable discretisation of the same underlying process. Hyperparameters then transfer effectively across batch-sizes, provided the "effective step size" remains small enough for the discretisation to remain valid.

**AdamW weight-decay reparameterisation**. We extend the parameterisation of Malladi et al. (2022), which yields a consistent SDE limit for Adam, to handle the weight-decay in AdamW. To informally motivate the weight-decay scaling rule, we consider the same simplifying RMSPropW example as in Malladi et al. (2022). We consider that the gradients queried at each step $k$ are a noisy version of a fixed direction $\boldsymbol{g}^k = \boldsymbol{g} + \sigma \boldsymbol{e}^k$, where $\boldsymbol{e}^k$ are i.i.d. Gaussian vectors of identity covariance. We note that as we move towards the small step size limit — i.e. as the batch-size gets smaller and smaller — the mini-batch gradient will be dominated by noise – $\sigma \gg \|\boldsymbol{g}\|$. Contrary to Malladi et al. (2022), we also consider a weight decay term as in AdamW. We denote $\eta$ the learning rate, and $\lambda$ the weight decay. We obtain the simplified RMSProp iterations (see (Malladi et al., 2022, Sec. 4.1) for more details) that define the iterates $\boldsymbol{\theta}(k; \eta, \lambda, \sigma)$ with the equation

$$\boldsymbol{\theta}^{k+1} = \boldsymbol{\theta}^k - \eta \left( \frac{\boldsymbol{g}^k}{\sigma} + \lambda \boldsymbol{\theta}^k \right) = \boldsymbol{\theta}^k - \frac{\eta}{\sigma} \left( \boldsymbol{g} + \lambda \sigma \boldsymbol{\theta}^k \right) - \eta \boldsymbol{e}^k, \tag{1}$$

which is a discretization of the SDE $d\Theta_t = \frac{1}{\eta\sigma}(\boldsymbol{g} + \lambda\sigma\Theta_t)dt + dW_t$ with step-size $\eta^2$, in the sense that $\boldsymbol{\theta}^k \simeq \Theta_{k\eta^2}$. Therefore, we find that the multipliers to keep fixed iterate distributions, i.e., such that $\boldsymbol{\theta}(k; \eta, \lambda, \sigma) = \boldsymbol{\theta}(m_k k; m_\eta \eta, m_\lambda \lambda, m_\sigma \sigma)$, should verify $m_k m_\eta^2 = 1$, $m_\eta m_\sigma = 1$, and $m_\lambda = m_\eta$. In particular, if the batch-size is multiplied by $\kappa$, we have $m_\sigma = \kappa^{-1/2}$ and we find that the new hyperparameters matching the SDE limit should follow the square-root scaling rule

$$\eta' = \sqrt{\kappa}\eta, \quad k' = \kappa k, \text{ and } \lambda' = \sqrt{\kappa}\lambda . \tag{2}$$

To the best of our knowledge, we are the first to extend the SDE reparametrisation scaling rules (Li et al., 2019; Malladi et al., 2022) to the weight decay of AdamW, although we note that the

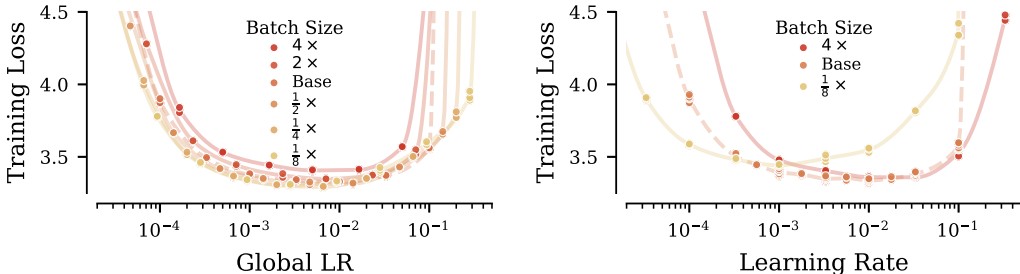

Figure 3: **Learning rate transfer with batch-size. Left:** Learning rates transfer when using the square-root rule in Equation (2) **Right:** Learning rates fail to transfer without adjustment. Each setting is run with three independent seeds.

same scaling rule for weight decay was proposed based on other principles in recent work (Wang & Aitchison, 2025; Compagnoni et al., 2025; Bergsma et al., 2025). We report the effect of using these scaling rules in batch-size in Figure 3; using the square-root rule is critical to good LR transfer. We further show in Figure 4 that the above rule is essential for transfer of weight-decay in batch-size.

**AdamLH and multipliers** Equation (1) mirrors the Pytorch implementation of AdamW, where the weight decay $\lambda$ is multiplied by the learning rate $\eta$. If one instead uses the original AdamW implementation, often coined AdamLH, as proposed by Loshchilov & Hutter (2017), we get the simplified iterations $\boldsymbol{\theta}^{k+1} = \boldsymbol{\theta}^k - \eta\frac{\boldsymbol{g}^k}{\sigma} - \lambda\boldsymbol{\theta}^k$, and we find that the multipliers are the same as for AdamW, except that $m_\lambda = m_\eta^2$: doubling the batch-size means that the weight decay should now be doubled. Hence, using AdamLH leads to a bigger drift across batch-sizes if the scaling is not done correctly, amplifying further the drift observed in Figure 3, right. We posit that this is one of the reasons why the Pytorch implementation is more widely used.

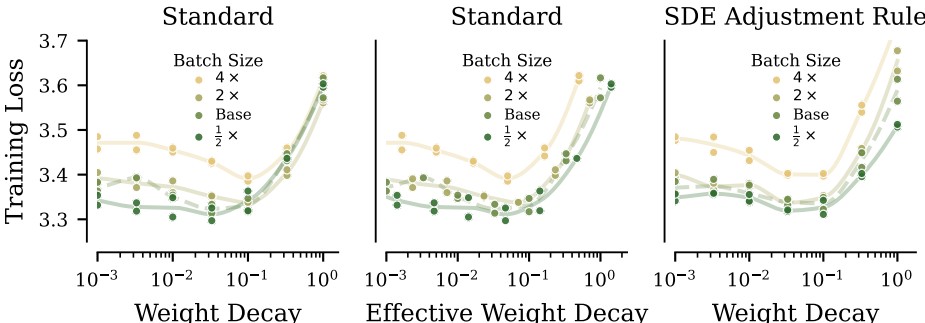

Figure 4: **Weight decay transfer with batch-size. Left:** Weight decay fails to transfer with batch-size without any adjustments. **Middle:** The rescaled (effective) weight decay $\lambda/\sqrt{\kappa}$ where $\kappa$ is the increase does transfer. **Right:** The effective weight decay transfers when rescaling all hyperparameters following our AdamW SDE scaling rule. Each setting is run with three independent seeds.

### 2.3 HYPERPARAMETER TRANSFER IN TRAINING DURATION

Unlike transfer in model size or batch-size, transfer in the token horizon has received comparatively less attention in the literature. Nonetheless, it is one of the two main levers to scaling compute. Like Bjorck et al. (2025), we observe that the optimal learning rate decays with the number of training iterations, holding all other things constant.

**Optimal learning rate decay with token horizon.** In Figure 5, we notice that the optimal learning rate decays at a rate roughly proportional to $\frac{1}{\sqrt{\kappa}}$, where $\kappa$ is the factor by which we've increased the number of training iterations. In Figure 5 *(right)*, we plot the optimal learning rate from the shortest training duration ($\eta_{\text{opt}}$) transferred with the scaling rule $\frac{\eta_{\text{opt}}}{\sqrt{\kappa}}$ in gray, and observe that it aligns almost perfectly with the true optima. In contrast, Bjorck et al. (2025) fit a scaling law to

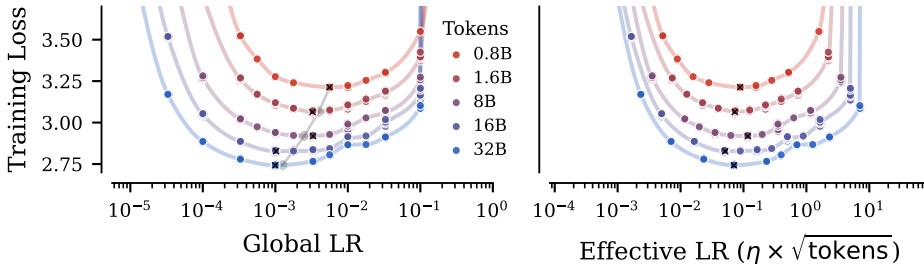

Figure 5: Learning rate transfer across training horizon – adjusting the number of tokens by changing the number of training iterations while holding batch-size constant. **Left:** Break-down of transfer of the global learning rate. **Right:** Stability of the "effective" learning rate – one that preserves the AdamW SDE integration horizon. Each setting is run with three independent seeds.

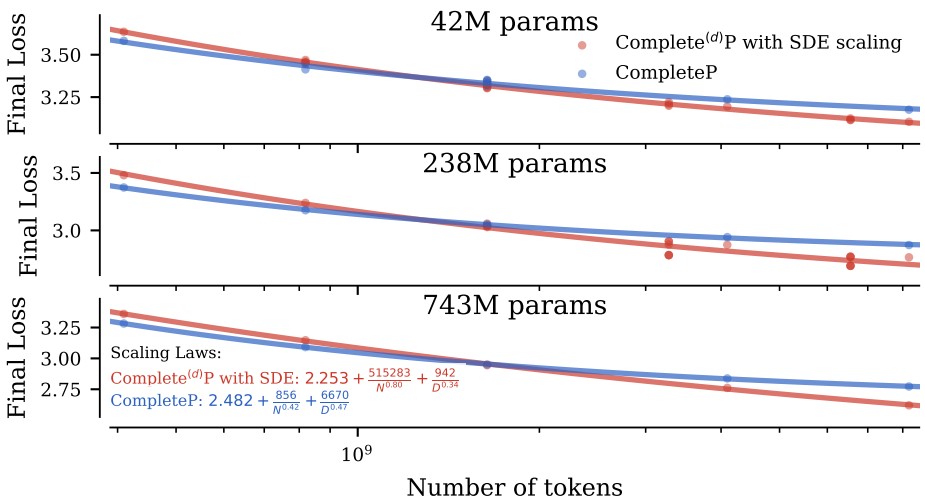

Figure 6: **Scaling law comparison of models trained with and without Complete^(d)^P token horizon scaling rule**. We compare Complete**(d)**P width & depth scaling only (aka CompleteP with $\epsilon$ and QK-norm fixes in Section 2.1) and full Complete**(d)**P with SDE scaling rules for token horizon transfer. The token horizon transfer rule leads to better performance at scale, as indicated by a better lower bound coefficient of the scaling law.

find the exponents $\beta$ for the scaling rule $\frac{\eta_{\text{opt}}}{\kappa^\beta}$; and they identify $\beta$ to be in the ranges of $0.3 - 0.7$ depending on the model size, which appears congruent with our scaling rule.

**SDE iso-horizon scaling rule**. In light of the SDE interpretation in Section 2.2, scaling the learning rate by $\frac{1}{\sqrt{\kappa}}$ while holding the batch-size constant can be seen as reducing the signal-to-noise (SNR) ratio in the SDE, while keeping the time horizon constant. Indeed, we orthogonally observe that when simulating the AdamW SDE, improving the signal-to-noise ratio (i.e. reducing the size of the diffusion coefficient) while holding other parameters constant consistently leads to improved performance. Hence, we hypothesise that the right way to scale the token horizon might be only adjusting the signal-to-noise parameter in the AdamW SDE, while keeping all other terms constant. We validate this hypothesis in Figure 14, where we scale the number of tokens by increasing batch-size only (which has the desired effect of changing the signal-to-noise ratio); we observe a near-perfect learning rate transfer across the token horizon. We further observe that **the resulting scaling rule as a function of the token horizon leads to better asymptotic performance, as predicted by a scaling law**, as we demonstrate in Figure 6.

The SDE iso-horizon scaling rule empirically permits transfer across training horizon.

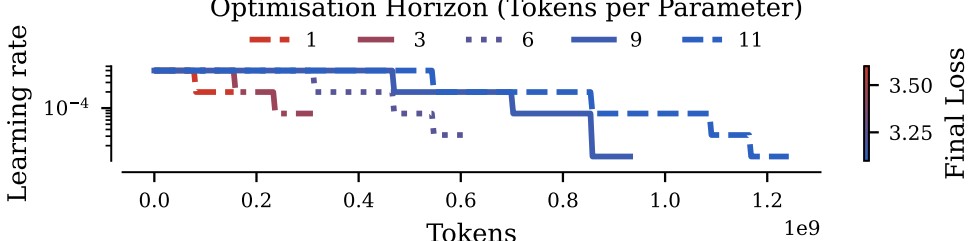

Figure 7: **Best Learning Rate annealing over 4,842 runs for five different token horizons**. The best schedule at a short horizon is never a prefix for the best schedule at a longer horizon. The optimal schedule cannot be found by a greedy approach: the best LR annealing is not data agnostic.

We expect this transfer to break at larger batch-sizes, where the discretisation will be too coarse for AdamW to approximate the underlying SDE, but when taken together with the batch-size reparameterisation rules in Section 2.2, this finding suggests how to scale all HPs across token horizons, while choosing the batch-size freely. This is the token horizon scaling procedure we follow in all the HP transfer results in the remainder of this paper, unless otherwise stated. We note that this finding might be specific to the fixed (cosine) schedule that we use.

**Best Learning Rate (LR) annealing at different token horizons.** Optimal schedules might have different shapes at different token horizons (Luo et al., 2025). We conduct a greedy search to determine optimal learning rate schedules in the following way. We enumerate all the non-increasing piecewise-constant LR schedule over the discrete set $\{0.0015/2.5^k | 0 \le k \le 4\}$. We sub-divide the total training duration in 16 intervals of 77M tokens each. At the end of each interval, either the LR remains constant, either it is decayed by one or more steps. For five different token horizons, we report the best scheduling among the 4,842 tested. We report the results in Figure 7. We notice that the best scheduling at short horizon is never a prefix of the best scheduling at long horizon. This empirical observation is compatible with the findings of Luo et al. (2025): there is a tension between the optimisation bias induced by the terminal LR value (the lower the better) and the progress of optimisation which requires higher LR values at start.

## 3 INVESTIGATING TRANSFER OF PER-MODULE HYPERPARAMETERS

Equipped with the tools for hyperparameter transfer described in the preceding section, in this section we investigate 1) how much there is to gain from per-parameter hyperparameter optimisation, and 2) how well do per-module hyperparameters transfer.

### 3.1 OPTIMISING PER-MODULE HYPERPARAMETERS

To show improvements and transfer of per-module hyperparameters, we need a good way to optimise them at a fixed scale. Although there is ample literature on hyperparameter optimisation in deep learning (Snoek et al., 2012; Maclaurin et al., 2015; Lorraine et al., 2020), optimising HPs on a per-module basis introduces many new difficulties. Below, we highlight why many standard approaches fail in the per-parameter optimisation setting.

**Per-module hyperparameter loss landscape** In Figure 8, we plot slices through the per-module learning rate (LR) loss landscape — i.e. the landscape of the mapping from LRs to the final loss. We observe that, fortuitously, it's pretty close to being invex (stationary points are global minima), and hence might be tractable even despite its high dimensionality. Several other aspects, however, render it challenging for common HP optimisation methods: **1)** The values of per-module learning rates at which training becomes unstable are module-dependent, and can differ by multiple orders of magnitude. **2)** The boundary at which training becomes unstable has a complex shape, with non-trivial interactions among different modules, implying it's difficult to predict with simple predictive models (e.g. linear models or Gaussian Processes (Williams & Rasmussen, 2006)). Our observation is similar to that made by Sohl-Dickstein (2024) — who observed the stable regime boundary

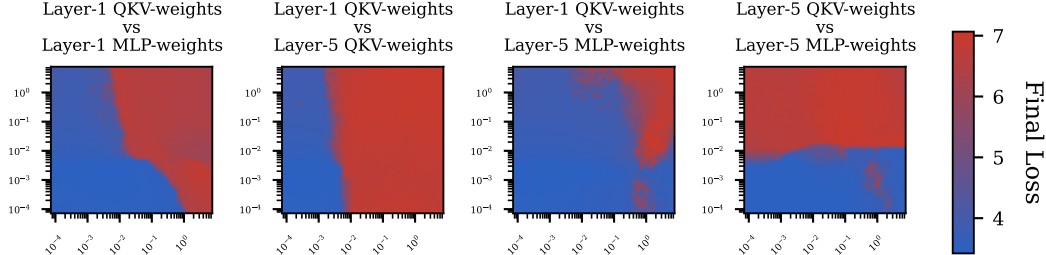

Figure 8: The boundary for stable training has a complex shape. Each plot shows the final training loss for different combination of learning rates for the modules indicated, while fixing the remaining learning rates to the optimal "global" value. MLP weights refer to the MLP in the attention layer. If unstable training results in NaNs, the last stable training loss is reported.

is a fractal — but we also note a lack of an emergent simple structure at the macro scale. This means common hyperparameter optimisation strategies, like random search or standard Bayesian Optimisation, struggle in this regime. For instance, random search lacks any locality bias; we observe that without careful manual tuning of the search boundaries, either all runs will fail due to unstable training, or the boundaries will fail to include the actual optimum. Commonly, Bayesian Optimisation relies on Gaussian Process (GP) models to guide search. However, GPs may struggle on highly non-stationary data (Snoek et al., 2014; Rana et al., 2017) with a fixed kernel. We did observe such irregularities in the per-module HP to loss landscape, which resulted in many failures when using BO. Many of these difficulties can be alleviated by more robust 'trust region' methods — approaches that optimise in neighbourhoods of previous good solutions. We describe a simple trust-region random search variant that we use for our experiments in Appendix C.

**Parameterising per-module hyperparameters** We adopt a depth–type Kronecker parameterisation of per-module hyperparameters that is compatible with Complete$^{(d)}$P transfer across width and depth. Let $m \in \mathcal{M}$ index module type within a Transformer block (QKV weights, attention projections, MLP weights and biases, layer norm and QK-norm multipliers), and let $\ell \in 1, \ldots, L$ index the depth. For a hyperparameter $\zeta_{g,\ell} \in \eta, \lambda, (1 - \beta_1), (1 - \beta_2), \epsilon$ for module $m$ at depth $\ell$:

$$\log \zeta_{m,\ell}(T,N,L,B) = \underbrace{\log \zeta_m^{\texttt{type}}}_{\text{type}} + \underbrace{\log \zeta_\ell^{\texttt{depth}}(L)}_{\text{depth}} + \underbrace{\log \text{SDE}(T,B)}_{\text{SDE batch/horizon}} + \underbrace{\log \text{CP}_m(N,L)}_{\text{CompleteP*}} \quad (3)$$

where $\text{SDE}(T; B)$ carries all training-horizon $T$ and batch-size $B$ dependence via the AdamW SDE transfer rules, $\text{CP}_g(N, L)$ is the Complete$^{(\mathbf{d})}$P scaling rule adjustment in width and depth (Table 1), and $\zeta_m^{\texttt{type}}$, $\zeta_\ell^{\texttt{depth}}$ for $m \in \mathcal{M}, \ell \in L$ are dimensionless, time-invariant multipliers that we optimise at a given width and depth. This factorisation reduces the number of free multipliers from $|\mathcal{M}|L$ to $|\mathcal{M}| + L$. We optimise the hyperparameter multipliers in log-space using trust-region random search (Appendix C). For the learning rate, the above multiplier post-multiplies the learning rate from the cosine schedule. When transferring the depth multipliers $\zeta_\ell^{\texttt{depth}}(L)$ to a larger depth $L'$, we interpolate them linearly with respect to $\frac{\ell}{L}$. This is reasonable, as the depth-SDE ($\alpha = \frac{1}{2}$) or depth-ODE limits ($\alpha \in (\frac{1}{2}, 1]$) should still exist if the base hyperparameters $\zeta_{\lfloor tL \rfloor}^{\texttt{depth}}(L)$ vary continuously with sufficient regularity across depth $t \in [0, 1]$. In this sense, the finite-depth multipliers can be seen as a discretisation of the continuous limit HPs: To transfer to a large depth we simply linearly interpolate all HPs in depth.

## 3.2 PER-MODULE HYPERPARAMETERS MATTER

**Depth multipliers matter** We ablated away the effect of the learning rate per-depth multipliers, by instead considering only a search over learning rate multipliers for each layer *role*: Within each residual block, every parameter group gets an independent learning rate, which is shared *across* different residual blocks; similarly, each parameter group in the embedding and unembedding layers gets its own value. We initiate this search from a projection of the best per-module hyperparameters onto this linear subspace. In Figure 16c, we observe that the search value, although still substantially better than the best global learning rate, is worse than the one that includes per-depth multipliers.

Hence, while **the majority of the gain comes from different module types within residual blocks getting different learning rates**, there is still notable benefit to per-depth multipliers.

**How restrictive is the depth-Kronecker factorisation?** To check how much performance we're leaving on the table with the depth-Kronecker factorisation constraint, we continue searching for fully uncoupled per-layer learning rates from the optimal Kronecker-factorised ones. The search results are shown in Figure 16b. Crucially, we observe virtually no improvement over the Kronecker-factorised ones. While this is not conclusive evidence that the optimal per-module learning rates are depth-Kronecker factored – the fully uncoupled search-space is much higher-dimensional and more difficult to navigate, and its likely we didn't find the optimum – these runs imply that most of the benefits of per-module HP optimisation can be captured by Kronecker factorised learning rates.

### 3.3 OPTIMAL PER-MODULE HYPERPARAMETERS TRANSFER WITH SCALE

Demonstrating upsides of per-module HP optimisation would be of little practical use if the HPs have to be tuned at the target model scale. In this section, we show that the improvements *do* persist across different model scales. Firstly, we demonstrate transfer in *model size*. Figure 11 illustrates that the optimal per-module learning rates transfer as we scale up both width and depth. Although we cannot easily visualise how the per-parameter HP loss landscape shifts as we vary model size (like was shown in Figure 2) due to its high-dimensional nature, we instead show the final training losses for a slice (a hyperplane) going through both the scaled-up optimal per-module learning rates and the optimal global learning rate. We observe that this landscape appears stable with model size, suggesting that the optimal per-module LRs do transfer with width and depth.

**Speed-ups from per-module HPs at large scale.** We also investigate what improvements are possible when transferring per-module HPs to a compute-optimal model at a larger scale. We jointly optimise the per-module learning rate, weight-decay, AdamW $\beta_1, \beta_2, \varepsilon$ and initialisation scale, and the residual block multipliers at the 50M parameter/1.6B token scale. We initiate the search from the optimal global HPs. In Figure 1, we show that the transferred per-module HPs lead to a $1.32\times$ speed-up over the optimal global HPs in a $\sim 10000\times$ higher compute setting.

## 4 DISCUSSION & CONCLUSION

**Limitations & Future Work** Our study encompasses a broad set of ablations, but there are limits to our empirical investigations and resulting recommendations, some of which we highlight below:

- Although we identify improved per-module hyperparameters in a reasonable number of trials, more trial-efficient search methods seem achievable. Exploring Bayesian Optimisation methods that address the specific difficulties of the per-module hyperparameter loss landscape highlighted in this work (e.g. Trust Region Bayesian Optimisation (Eriksson et al., 2019)), utilise early-stopping, and exploit the structure of the training process (Lin et al., 2024), seem particularly promising.
- We only evaluate one training setup (autoregressive transformer training on the RedPajama dataset). While that setup has broad practical relevance, our approach (the proposed transfer principles, especially for token horizon transfer, and benefits of per-module hyperparameters) should ideally be verified in other settings.
- We only consider one fixed learning rate schedule (cosine decay) throughout. The optimal *schedule* for each token horizon might be very different to the recommendation in this work, and might improve upon a fixed schedule.
- The speed-ups of per-module hyperparameters we identified at a small scale seem to diminish slowly with model and data size. However, we do not know whether that is mostly to be explained by imperfect hyperparameter transfer in the non-asymptotic regime, or whether this is due to an asymptotic property of the infinite-scale models. We hope future work might find computationally feasible ways of answering that question.

**Conclusion** In this paper, we proposed new transfer rules for hyper-parameters, valid across the most important scaling axes: model's width, model's depth, token horizon, and batch size. Furthermore, these transfer rules also hold for *per-module* hyper-parameters. We demonstrate that systematic optimisation at small scale with trust region methods produce a configuration that transfers to larger scales, and significantly improves training speed.

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

## CONTENTS

# A ADDITIONAL EXPERIMENTS & FIGURES

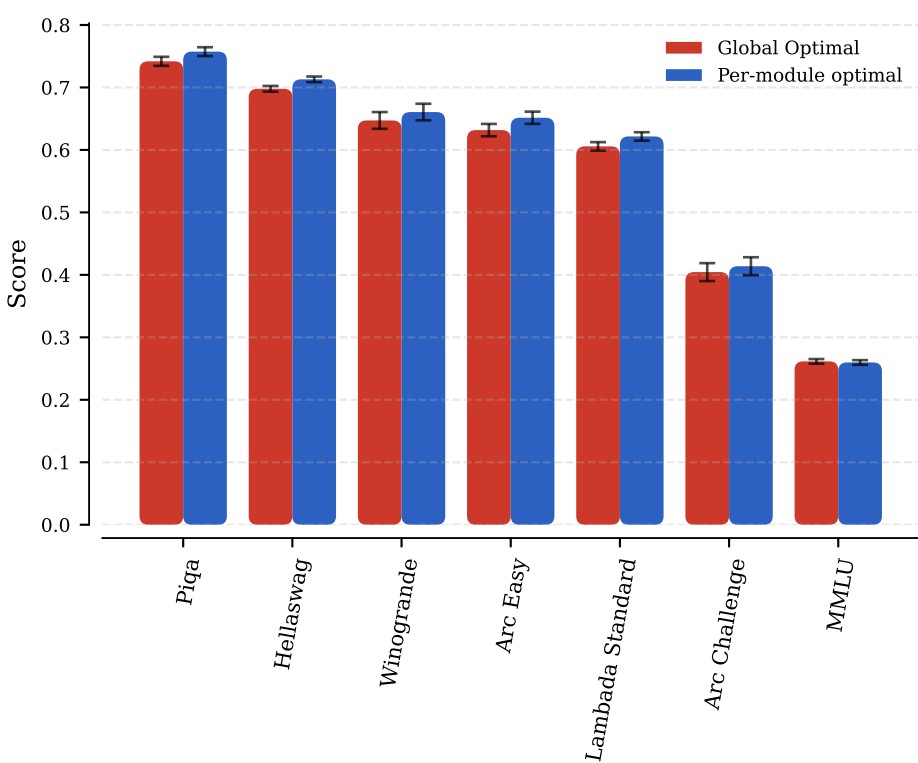

Figure 9: Final down-stream benchmark performance for the models trained with the optimal per-module hyperparameters and the optimal global hyperparameters, both transferred with Complete[(d)]P , for the 7.2B parameter/143B token training runs from Figure 1.

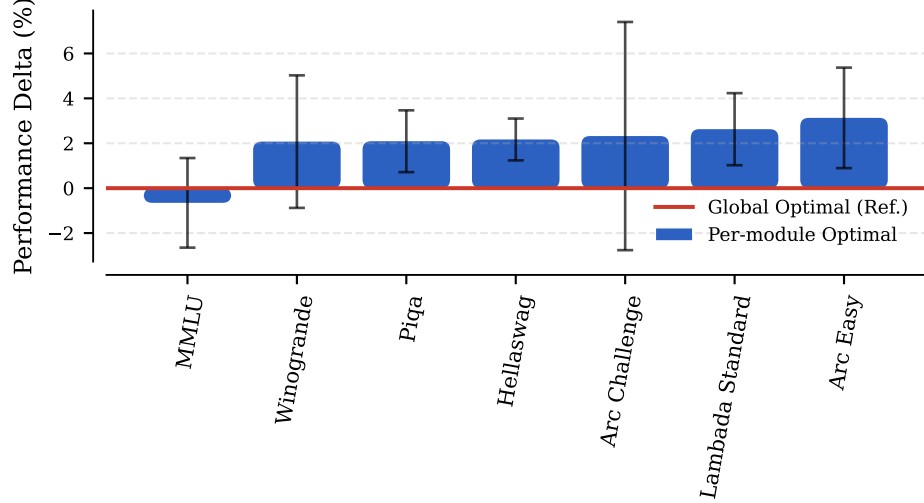

Figure 10: Final down-stream benchmark performance for the model trained with the optimal per-module hyperparameters for the 7.2B parameter/143B token training runs from Figure 1.

## A.1 SUBOPTIMALITY OF LEARNING RATES IDENTIFIED AT TOO SMALL SCALES

As hyperparameter transfer in width & depth with Complete[(d)]P is motivated by the asymptotic width & depth behavior, one would expect it to start degrading at smaller scales. Indeed, this can

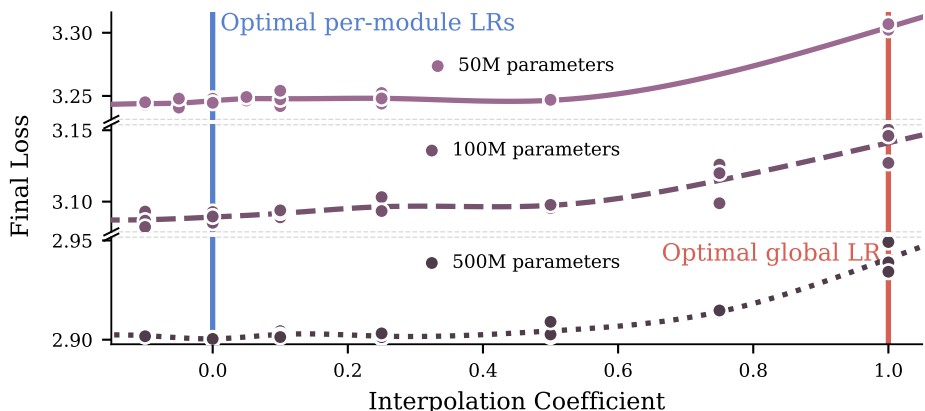

Figure 11: Transfer across model scale of the optimal per-module learning rates. We interpolate between the optimal global learning rate multiplier, and the optimal per-module multipliers across models of different scales, and show that the optimal per-module multipliers **1)** consistently improve upon the global multiplier baseline, and **2)** remain close to optimal in the hyperplane spanned by the optimal global and local multipliers. Each setting is run with three independent seeds.

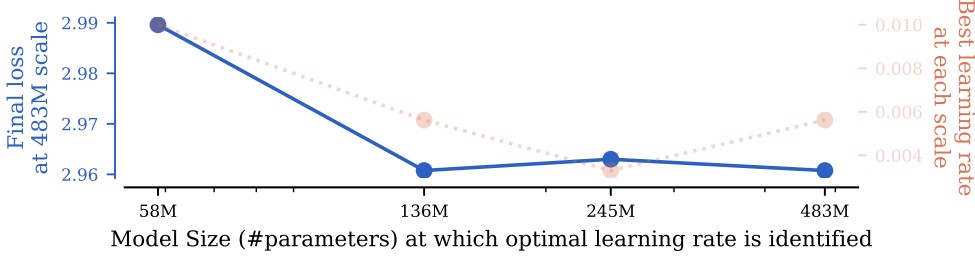

Figure 12: **Suboptimality of learning rates optimised at smaller scales.** We plot the final loss on a 483M parameter model (*y-axis*) when training with the learning optimal for a smaller model (*x-axis*). To optimise the (global) learning rate at each of the shown smaller scales, we conduct a grid search with the following set of candidates: $\{10^{-4}, 3.3\times, 10^{-4}, 5.6\times10^{-4}, 10^{-3}, 3.3\times, 10^{-3}, 5.6\times 10^{-3}, 10^{-2}, 3.3\times, 10^{-2}, 10^{-1}, 3.3\times, 10^{-1}\}$. We use Complete[d]P throughout. At the 136M scale, the optimal learning rate is already approximately stabilised. At the 50M scale, we incur a small penalty. The final losses are averages over 3 seeds.

be empirically observed with learning rate transfer in, e.g., Figure 2. Hence, there is a trade-off when optimising hyperparameters with Complete[d]P transfer; going to smaller model sizes enables cheaper hyperparameter optimisation, but these hyperparameters could be slightly suboptimal at scale due to degraded hyperparameter transfer.

We illustrate this trade-off in Figure 12, where we show the final loss of a larger-scale model (483M) when transferring optimal hyperparameters (global learning rate) from a smaller model with Complete[d]P . We see that when transferring the optimal learning rate from a smaller 58M model, we incur a small transfer penalty. Optimising the learning rate at the 136M scale or larger seems to incur virtually no penalty. This suggests we might been able to obtain more competitive per-module hyperparameters, at a substantially higher compute cost, were we to conduct our search at the 136M scale. We consider it an exciting direction for future work to empirically investigate at what scales hyperparameters should be optimised, and subsequently transferred, for maximum compute savings. Nonetheless, this will of course depend on the hyperparameter search method used.

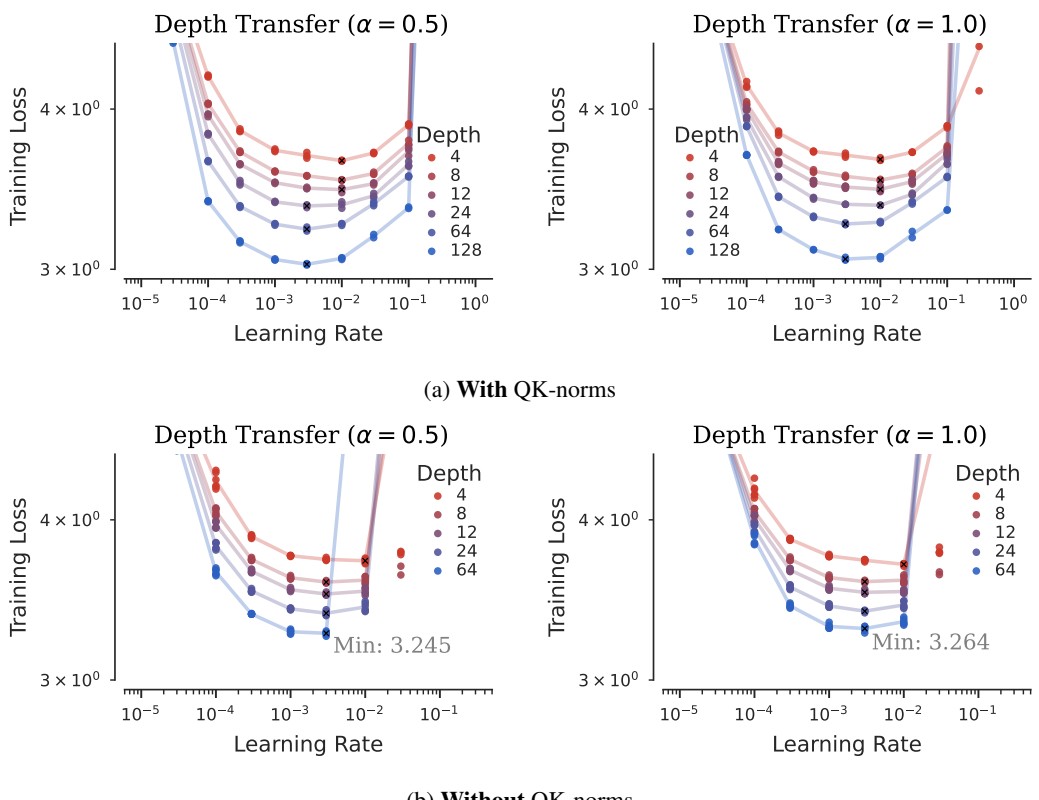

(a) **With** QK-norms

(b) **Without** QK-norms

Figure 13: The effect of QK-norms on hyperparameter transfer of the global learning rate across depth with two variants of Complete$^{(d)}$P with $\alpha \in \{\frac{1}{2}, 1\}$.

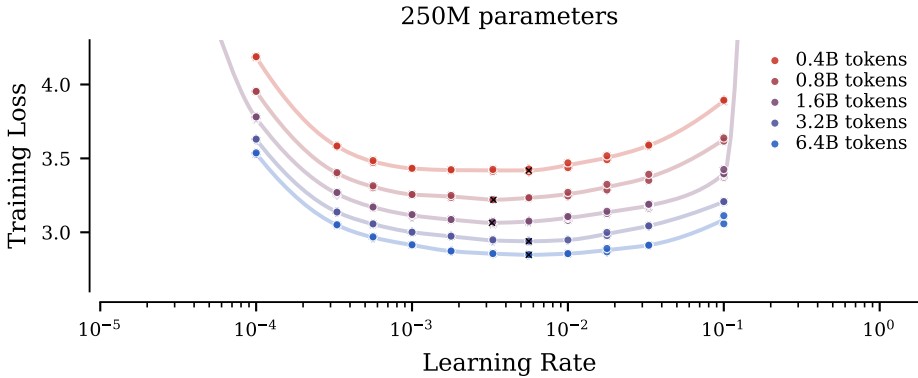

Figure 14: Learning rate transfer across token horizon when scaling up by increasing batch-size while holding training iterations constant. This scaling rule can be seen as improving the gradient signal-to-noise (SNR) ratio in the discretised AdamW SDE (Malladi et al., 2022), while holding all the other SDE parameters and the integration horizon fixed.

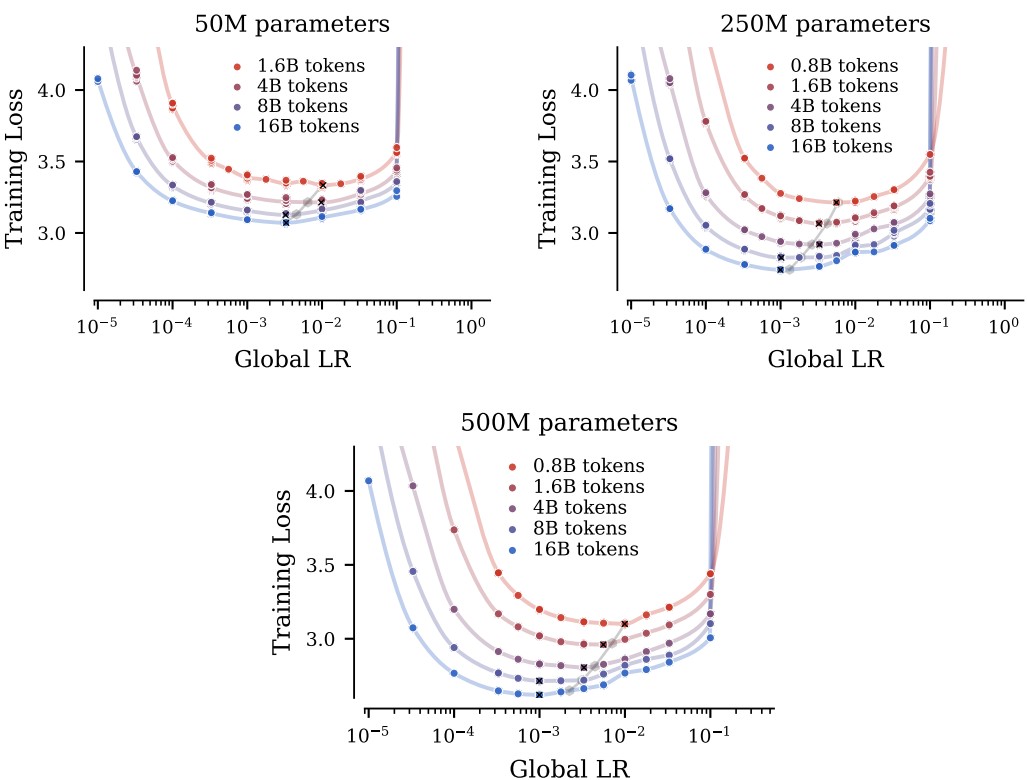

Figure 15: Lack of learning rate transfer across training horizons — increasing token horizon through number of iterations with a fixed batch-size — for different model sizes. The square-root transfer rule for the optimal learning rate identified at the smallest token horizon for each model size is plotted in .

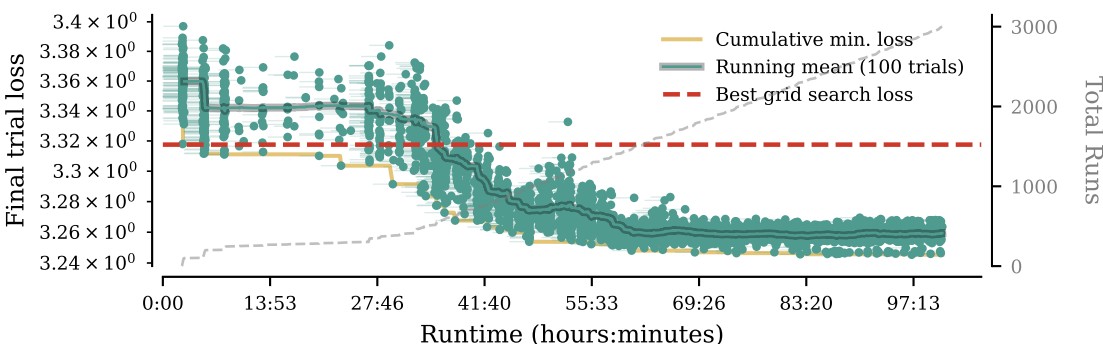

(a) Hyperparameter search for the per-module learning rate multipliers parameterised with the depth-Kronecker factorisation.

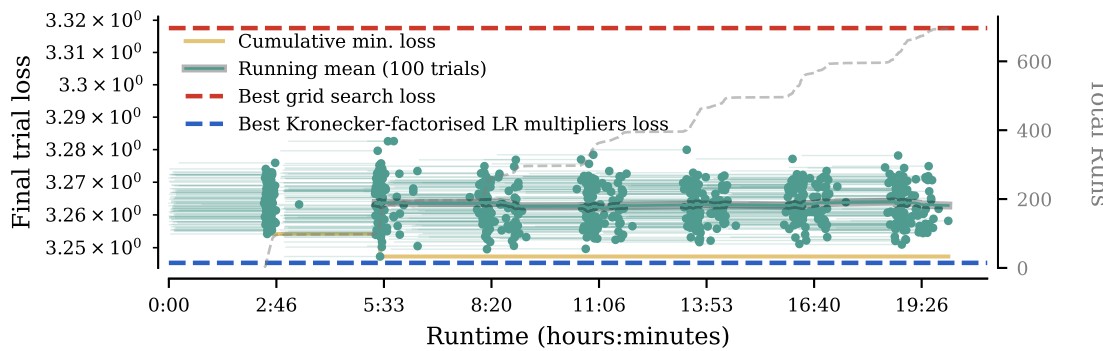

(b) Hyperparameter search for the per-module learning rate multipliers with **fully uncoupled** multipliers. The search is initialised with the optimal HPs found in the search for optimal depth-Kronecker factorised multipliers in Figure 16a.

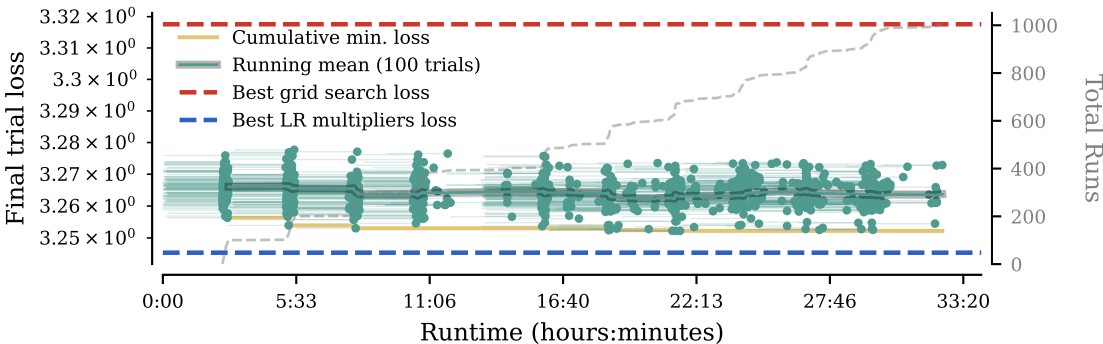

(c) Hyperparameter search for the per-module learning rate multipliers **with no depth multipliers**. he search is initialised with the projection of onto the constraint set of the optimal HPs found in the search in Figure 16a.

Figure 16: Hyperparameter search results with Trust Region Random Search. Each dot indicates the final loss of a single trial, and the lines indicate the training duration (start & end).

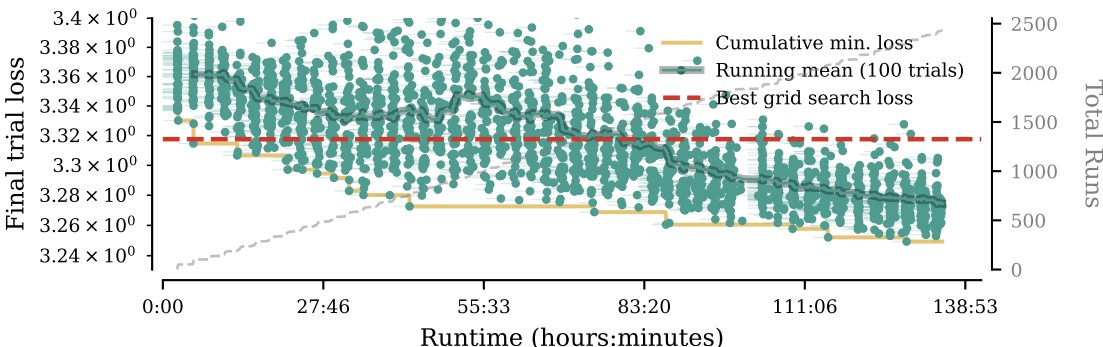

Figure 17: Hyperparameter search for the per-module learning rate multipliers parameterised with the Kronecker factorisation with CMA-ES. C.f. Figure 16a for the comparable Trust-region Random Search results.

# B MOTIVATION OF THE COMPLETE$^{(D)}$P ADJUSTMENTS

Here, we give a justification for each of the modifications made in Complete$^{(d)}$P in Table 1. These modifications primarily concern scaling the infinite-width limit. We directly rely on the properties that $\mu$P parameterised neural networks are known to possess that were formally shown in (Yang et al., 2024; Yang & Littwin, 2023). Concretetely, we note that when scaling with $\mu$P, all forward hidden-layer (pre-)activations are expected to have entries of size $\Theta(1)$ (as defined in (Yang & Hu, 2021, Definition N.2)) and the backpropagated gradients with respect to hidden (pre-)activations are expected to have entries of size $\Theta(1/N)$. Furthermore, the (pre-)activations and the back-propagated (pre-)activation gradients are expected to approach *i.i.d.* in the infinite-width limit.

## B.1 QK NORM MULTIPLIER WEIGHTS

In standard implementation of QK norms, the elementwise affine operation $\mathbf{x} \mapsto \mathbf{m} \odot \mathbf{x} + \mathbf{b}$ with multipliers $\mathbf{m}$ and bias $\mathbf{b}$ is shared across the transformer heads. When scaling width by increasing the number of heads — as is common in most of the relevant model scale parameterisation literature (Dey et al., 2025; Yang et al., 2022) — this effectively means that these parameters are shared across the scaled width dimension $N$. For instance, for a collection of query vectors $\mathbf{q} \in \mathbb{R}^{N_{\text{heads}} \times d_{\text{head}}}$, we have that the normalised query elements $\hat{q}_{ij} := m_j q_{i,j} + b_j$ all share the same parameters $m_j, b_j \in \mathbb{R}$ for $i = 1, \ldots, N_{\text{heads}}$, where $N_{\text{heads}} = \Theta(N)$. We denote by $\mathbf{q}_{:,j}$ the $\mathbb{R}^{N_{\text{heads}}}$ vector $(q_{i,j} : i = 1, \ldots, N_{\text{heads}}$ ($\hat{\mathbf{q}}_{:,j}$ respectively). By the results of Yang & Hu (2021); Yang & Littwin (2023), we have that for the $\mu$P parameterisation $\mathbf{q}_{:,j}$ has entries of size $\Theta(1)$ throughout training. The loss gradients for any hidden (pre-)activation, are known to have entry size $\Theta(1/N)$, and so the post-normalised query activation gradients $\frac{\partial \mathcal{L}}{\partial \hat{\mathbf{q}}_{:,j}}$ will also be of size $\Theta(1/N)$. The backpropagated gradient with respect to the multiplier $m_j$ is:

$$\sum_{i=1}^{N_{\text{heads}}} \left[ \frac{\partial \mathcal{L}}{\partial \hat{\mathbf{q}}_{:,j}} \right]_i \mathbf{q}_i = \frac{1}{N} \sum_{i=1}^{N_{\text{heads}}} N \left[ \frac{\partial \mathcal{L}}{\partial \hat{\mathbf{q}}_{:,j}} \right]_i \mathbf{q}_i,$$

where the rescaled random variables $N \left[ \frac{\partial \mathcal{L}}{\partial \hat{\mathbf{q}}_{:,j}} \right]_i \mathbf{q}_i$ have entry size $\Theta(1)$ as $N \to \infty$. Informally, in Yang & Hu (2021), the random variables $N \left[ \frac{\partial \mathcal{L}}{\partial \hat{\mathbf{q}}_{:,j}} \right]_i \mathbf{q}_i$ for $i = 1, \ldots, N_{\text{heads}}$ were shown to approach *i.i.d.* as $N \to \infty$. Hence the sum above has a Strong Law of Large Numbers like behaviour, converging to the mean of the entrywise limit of $N \left[ \frac{\partial \mathcal{L}}{\partial \hat{\mathbf{q}}_{:,j}} \right]_i \mathbf{q}_i$. As such, we effectively have that the gradient with respect to the width-shared parameters is also $\Theta(1)$ with width. The scale of the AdamW $\epsilon$ should match the scale of the gradient (Yang & Littwin, 2023), and so we have that the AdamW $\epsilon$ parameter for the width-shared multipliers should also be scaled as $\Theta(1)$ with width. A near-identical argument follows for the bias terms.

## B.2 EMBEDDING LAYER ADAMW $\epsilon$

The $\Theta(1/N)$ scaling with width $N$ for the embedding layer AdamW $\epsilon$ follows from the observation that the gradients with respect to the embedding parameters have element size $\Theta(1/N)$. To see this, note that the gradients with respect to the output of the embedding layer are $\Theta(1/N)$, whereas inputs are obviously constant with width. Hence, it naturally follows that AdamW $\epsilon$ should be scaled as $\Theta(1/N)$ to match the scale of the gradient (Yang & Littwin, 2023).

## B.3 CHANGES TO THE UNEMBEDDING WEIGHT

The changes to the unembedding weight scaling rules are mostly a reparameterisation of the multiplier-based $\mu$P implementation in (Dey et al., 2025). Namely, for AdamW, a weight multiplier $m_N^\gamma$ has the same effect throughout training (bar the finite-precision arithmetic effects) as: **1)** multiplying the initialisation variance by $m_N^{2\gamma}$, **2)** multiplying the learning rate by $m_N^\gamma$, **3)** and multiplying the AdamW $\epsilon$ parameter by $m_N$ (Yang & Littwin, 2023). We re-parameterise with **(1)** and **(2)**, but we don't change AdamW $\epsilon$ as it appears to have been derived incorrectly in (Dey et al., 2025). To see this, note that with $\mu$P (the Table 3 variant without an output layer multiplier), the

gradients for the unembedding layer weights are expected to have scale $\Theta(1)$ with width $N$. Hence, to remain of the same scale, the output embedding weight $\epsilon$ should also have a matching scale of $\Theta(1)$ (Yang & Littwin, 2023). After reparameterisation to a $m_N^{-1}$ output layer multiplier — as is done in CompleteP — the $\epsilon$ would also have to had to be scaled as $m_N^{-1}$ to match the reparameterised gradients.

## C  PER-MODULE HYPERPARAMETER SEARCH ALGORITHM

As described in Section 3.1, standard random search is unsuitable for the task of optimising *per-module* hyperparameters. We make two minimal tweaks that make it into a workable method. We induce an exploitation bias by turning it into a trust region method: we constrain the search-space adaptively to the neighbourhood $\{\mathbf{x} \in \mathbb{R}^d : \|\mathbf{x} - \mathbf{x}_t^{\mathrm{opt}}\|_\infty \leq r\}$ of the current best solution $\mathbf{x}_t^{\mathrm{opt}}$ at a given iteration $t$. Hence, the bounds move with the best solution found so far. We optimise all parameter in the $\log_2$-space, and sample uniformly from within the bounding box. Even with this modification, however, we found that this trust-region random search quickly plateaued with a relatively high variance in the final loss values. Hence, to allow the algorithm to explore promising regions more thoroughly, we also decay the size of the bounding region $r$ if the loss doesn't improve after a certain number of trials.

For all experiments, unless stated otherwise, we instantiate the search with the bounding box size of 1 (meaning that at each iteration, we multiply the best solution found so far by $2^x$ with $x$ sampled uniformly from $[-1, 1]$), and decay size of the trust region $r$ by $0.7$ if no improvement is observed in 100 trials. We run the algorithm asynchronously with a maximum of 100 simultaneous trials.

The goal of this paper is not to identify the *best* HP optimisation strategy for this setting; we merely want to find a workable one in order to demonstrate potential for improvements from per-module HP search. Since the above tweaks borrow from the principles underlying many evolutionary search (ES) methods, we also wanted to directly compare to a strong ES baseline to check our method performs reasonably. In Figure 17, we compare CMA Evolutionary Search (CMA-ES) (Hansen, 2006) to the Trust-region Random Search described above (c.f. Figure 16a). CMA-ES is not natively an asynchronous HP search strategy, so we make a minor modification: for a population size $P$, we wait until at least $P$ trials sampled from the current generation have finished running. At that point, there might be more than $P$ new finished trials (left-over trials from the previous generations), so we update the CMA-ES state with $P$ *best* trials only. In this instance, Trust-region Random Search outperforms this CMA-ES variant. This gives credence to our search method of choice being able to identify good per-module HPs in reasonable runtime. We hope that future work can explore alternative strategies that might be able to severely reduce the number of trials required to find good per-module HPs.

## D  EXPERIMENTAL DETAILS

### D.1  BEST LEARNING RATE (LR) ANNEALING AT DIFFERENT TOKEN HORIZONS

We pretrain a small GPT-2 model (121M parameters). We enumerate all the non-increasing piecewise-constant LR schedule over the discrete set $\{0.0015/2.5^k | 0 \leq k \leq k_{max}\}$. We sub-divide the total training duration in $L$ intervals of 77M tokens each. At the end of each interval, either the LR remains constant, either it is decayed by one or more steps. We chose $L = 16$ and $k_{max} = 4$, which yields a total of 4842 runs. For efficiency, we use the same checkpoint to warm start all runs sharing the same prefix in the LR scheduling, which cut down the computational complexity of this naive enumeration from $\mathcal{O}(L^{k_{max}+1})$ to $\mathcal{O}(L^{k_{max}})$. Therefore, the total compute budget is kept under 7,000 A100 GPUh. For five different token horizons (155M, 310M, 621M, 932M and 1.24B) we report the best scheduling among the 4,842 tested. We report the results in Figure 7. We notice that the best scheduling at short horizon is never a prefix of the best scheduling at long horizon. This empirical observation is compatible with the findings of Luo et al. (2025): there is a tension between the optimisation bias induced by the terminal LR value (the lower the better) and the progress of optimisation which requires higher LR at start.

## D.2 BASELINE GLOBAL HYPERPARAMETER TUNING

To establish a baseline, we perform an extensive random hyperparameter search consisting of 2048 trials. Each trial trains a 50M parameter model ($d_{\text{model}} = 512, L = 4$) for 1.64B tokens (33 tokens/parameter) over a discrete search space defined by:

- LR $\in \{1 \times 10^{-4}, 3 \times 10^{-4}, 4 \times 10^{-4}, 1 \times 10^{-3}, 3 \times 10^{-3}, 4 \times 10^{-3}, 1 \times 10^{-2}, 2 \times 10^{-2}, 3 \times 10^{-2}, 1 \times 10^{-1}\}$
- Adam $\epsilon \in \{1 \times 10^{-14}, 1 \times 10^{-12}, 1 \times 10^{-10}, 1 \times 10^{-8}, 3 \times 10^{-8}, 4 \times 10^{-8}, 1 \times 10^{-7}\}$
- Adam $\beta_1 \in \{0.8, 0.85, 0.9, 0.95, 0.999\}$
- Adam $\beta_2 \in \{0.9, 0.95, 0.98, 0.99, 0.999\}$
- Weight Decay $\in \{1 \times 10^{-4}, 1 \times 10^{-3}, 1 \times 10^{-2}, 1 \times 10^{-1}, 2 \times 10^{-1}, 4 \times 10^{-1}\}$

The results and hyperparameter sensitivities from this search are visualized in Figures 18b to 18d. The optimal configuration from this global search achieves a validation negative log-likelihood of 3.34 nats. This result is substantially higher than that achieved by our per-parameter search strategy, underscoring the advantage of discovering optimal configurations at a small scale before upscaling with principled rules like Complete[(d)]P .

## D.3 PER-MODULE HYPERPARAMETER SEARCH AND TRANSFER (FIGURE 1)

**Per-module hyperparameter search** For the per-module hyperparameter search, we ran the trust-region random search as described in Appendix C. We ran 100 trials (training runs) in parallel, with a total budget of 5000 trials. We randomly chose a different random seed (dictating the network initialisation and data order) for each trial.

We optimised AdamW learning rate, weight-decay, $\epsilon$, momenta $\alpha_1 := (1 - \beta_1)$ and $\alpha_2 := (1 - \beta_2)$, as well as the standard deviation for the initialisation, with one hyperparameter (multiplier) per module type. For module types, we treat each 'tensor' within a transformer block as an individual type (e.g. QK-norm multipliers, QKV weights, output projection weights, first feedforward layer weights, second feedforward layer weights, etc. would all be individual types); each tensor outside the transformer blocks are also individual module types (input embedding weight, output embedding weight, output embedding bias, output layer norm multipliers, etc. would all be individual types). We also optimise the per-depth transformer residual block multipliers in the depth-type Kronecker parameterisation (there are two residual multipliers in each transformer block – one for the attention block, one for the feedforward block). Altogether, that leads to 79 hyperparameters to optimise.

The hyperparameter search at small scale in Figure 1 took 6730 GPU-hours on NVIDIA A100s, although 99% of the loss gains over the optimal global hyperparameters were realised within the first 3168 GPU-hours.

# E EXTENDED RELATED WORK

**Hyperparameter transfer in width & depth**. Our work directly builds upon, extends and combines many existing parameterisation for transfer across different modalities. For width transfer, we directly build on the Tensor Programs (Yang & Hu, 2021; Yang et al., 2022) based derivations for the $\mu$-parameterisation, and the extensions to adaptive optimisers (Yang & Littwin, 2023). Although (Yang & Littwin, 2023) contains an exposition of all the theoretical tools required to derive the right parameterisation for virtually any neural network architecture, applying these tools is still a non-trivial task. Yang et al. (2024) extended similar principles to find parameterisations in depth. Dey et al. (2025) adapted these principles to derive a width & depth transfer-enabling parameterisation specifically for transformer models. Although Dey et al. (2025) directly builds upon and uses virtually the same principles as (Yang & Littwin, 2023) and Yang et al. (2024), they do derive the right parameterisation for a broad range of hyperparameters (initialisation scales, learning rates, AdamW weight decay, AdamW epsilon), unlike the original $\mu$P paper (Yang et al., 2022), and they derive a complete set of rules specifically for transformer models. We directly build upon and extend CompleteP (Dey et al., 2025) for width & depth transfer part of our parameterisation, making a couple of

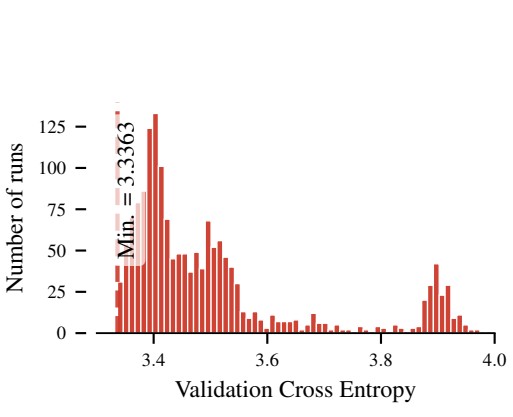

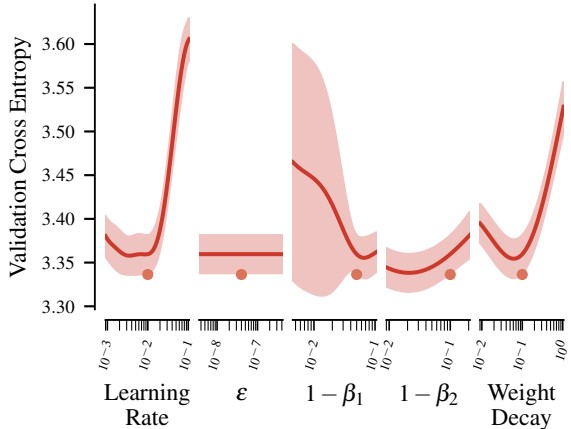

(a) Number of trials of baseline 50M model vs. evaluation loss.

(b) Gaussian Process fit sensitivity around optimal identified global hyperparameters.

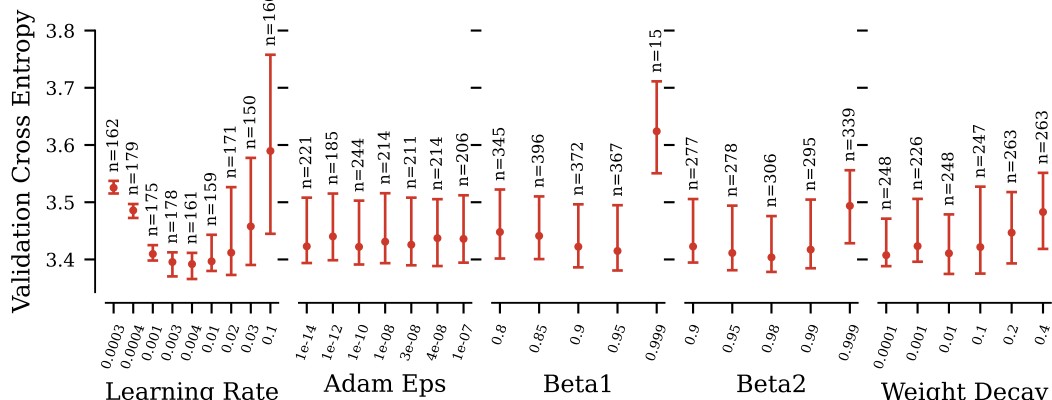

(c) (Marginal) sensitivity of the baseline 50M model for core hyperparameters. We show the 25%-75% quantiles for the validation losses with all hyperparameters sampled independently at random, conditioned on the shown hyperparameter being set to a certain value.

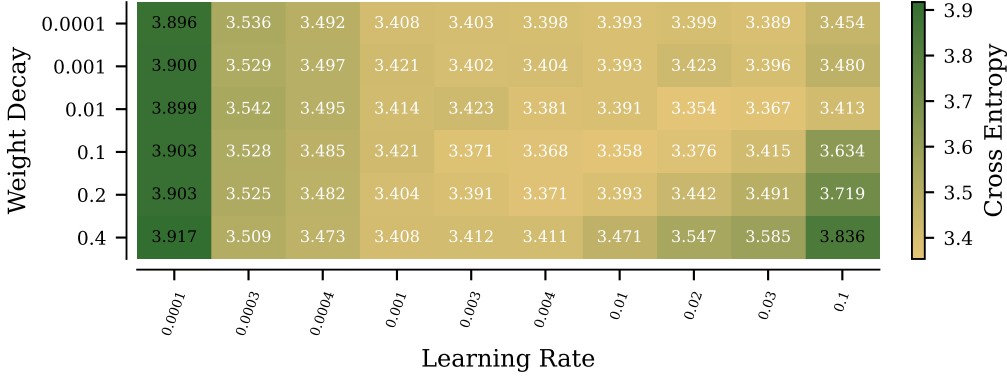

(d) Weight decay vs. learning rate for baseline 50M model.

Figure 18: Summary of the global hyperparameter sweep on a 50M parameter model for the baseline.

small modifications (extending to QK-norms and fixing minor mistakes in the derived rules)[2]. This constitutes one part of the Complete(d) parameterisation.

---

[2]One component that could, in principle, receive the same scaling treatment is gradient clipping. Gradient clipping can in principle be cast within the Tensor Programs framework, but would require the modification

**Transfer in batch-size and SDE scaling rules**. For the batch-size transfer rules for the Complete[(d)] parameterisation, we also directly build upon prior work on SDE transfer rules (Li et al., 2019), including for Adam (Malladi et al., 2022). We extend the SDEs of Malladi et al. (2022) to AdamW, allowing us to propose a scaling rule for weight-decay with batch-size. We note that similar scaling rules for weight-decay have been recently proposed in other works (Wang & Aitchison, 2025; Bergsma et al., 2025) prior to ours. To our knowledge, we are the first ones to motivate them theoretically with the principle of preserving the dynamics of an AdamW stochastic differential equation (SDE), integrating it with the transfer rules in batch-size for the other AdamW hyperparameters.

Our rules are compatible with those proposed by those proposed in (Wang & Aitchison, 2025; Bergsma et al., 2025). Wang & Aitchison (2025) posit how the product of weight-decay and learning rate should scale as a function of the batch-size. In particular, they suggest that the learning rate $\gamma(B)$ and the weight-decay $\lambda(B)$ should be scaled with batch-size $B$ so as to keep $\tau_{\text{EMA}} = \frac{B}{\gamma \lambda D}$ constant. This rule doesn't specify whether $\gamma$ or $\lambda$ should be adjusted, but only constrains their product. Substituting in our rules for $\gamma(B)$ and $\lambda(B)$ from Table 1 — which dictate that $\gamma(B) \propto \sqrt{B}$ and $\lambda(B) \propto \sqrt{B}$ — we see that $\tau_{\text{EMA}} \propto 1$. Hence, the rules we put forward are compatible with those of in Wang & Aitchison (2025). They are also more specific, dictating how the learning rate and weight-decay should each be adjusted individually.

Compagnoni et al. (2025) also propose an AdamW SDE, and suggest the same weight-decay scaling rule, but based on different principles. Whereas we argue for preserving the dynamics of the SDE (and, hence, approximately preserving the dynamics of discrete-time AdamW) similarly to Malladi et al. (2022), Compagnoni et al. (2025) propose scaling rules to try and maintain an upper bound on the final training loss. Furthermore, their derived SDE is *different* from ours and that of Malladi et al. (2022), whereas ours is fully compatible with that of Malladi et al. (2022). We highlight that Malladi et al. (2022) made a compelling case for the need for reparametrising the AdamW momenta terms that led to their SDE. This reparametrisation is missing from (Compagnoni et al., 2025).

**Token horizon transfer rules**. To the best of our knowledge, we are the first to propose our scaling rule in token horizon for learning rate, weight-decay, AdamW momenta and AdamW $\epsilon$ on the grounds of the proposed SDE principles. However, prior works have explored transfer in token horizon more broadly. (Bjorck et al., 2025) demonstrate that the optimal learning rate shifts with the token horizon, and propose an empirically derived scaling rule. Qiu et al. (2025) observe that *adequately normalised* training curves transfer across model size and token horizon when both are scaled jointly to remain compute optimal. Since they rescale by the final loss – the exact quantity the behaviour of which we study – it's not immediately clear how those insights could be exploited for hyperparameter transfer. Nonetheless, their observations might be closely related, and their analysis through the lens of SDEs could prove to be a fruitful avenue for explaining the transfer rules across token horizons we identify in this paper.

**Per-module hyperparameter selection**. Independently, (Ludziejewski et al., 2025) studied the benefits of setting hyperparameters differently for different parameter groups. They similarly show improvements over global hyperparameters at a fixed model scale, and demonstrate improvements persist when training a larger mixture of experts model ($8\times$ compute) with each expert being the same size as the base model. Our analysis differs in a few places: **1)** we thoroughly investigate transfer across model scale for the same architecture with $\mu$P and $\mu$P-derived parameterisations, **2)** we investigate transfer across token horizons and batch-size, **3)** we investigate more fine-grained hyperparameter transfer on a per-module basis, whereas Ludziejewski et al. (2025) only consider separating the parameters into broad groups, and **4)** we investigate hyperparameters beyond the learning rates (weight decay, AdamW momenta, etc.), whereas Ludziejewski et al. (2025) consider learning rates and parameters of their schedules.

Recently, Wang et al. (2025) motivate why per-module learning rates might be beneficial from a curvature perspective. They consider the relative scale for the **learning rates** for 5 sub-modules in a transformer block; more specifically, the query-key (QK) and value-output (VO) blocks; point-wise feedforward networks (FFN); normalization layers (Norm), and embedding layers (Emb). Following their theoretical analysis, their practical recommendation is to tune manually (informed by that theory) these ratios on a smaller scale and re-used as is for larger scales (which would still require

---

to how the norm is computed – the norm of each tensor ought to be scaled differently depending on the width before aggregating.

tuning a global LR when changing the compute scale). Compared to our work, their practical approach is restricted to a far smaller set of HPs: 5 LRs in their work vs. approximately a hundred HP in our case, since we study 6 fundamental HPs (learning rate, initialization scale, Adam $\varepsilon$, $\beta_1$, $\beta_2$ and weight decay) times (number of modules plus depth). Additionally, their perspective does not stress transfer of these hyperparameters along any scaling axes, which is a core contribution in our work. Instead, they retune the only hyperparameter they tune multipliers for (the learning rate) at *every* scale they consider, making it costly to apply it at scale.

