# OpenReview forum: "Completed Hyperparameter Transfer across Modules, Width, Depth, Batch and Duration"
_ICLR.cc/2026/Conference — ICLR 2026 Poster_

### Official Review · Reviewer_36xD · 2025-10-31

**Soundness:** 4
**Presentation:** 3
**Contribution:** 3
**Rating:** 8
**Confidence:** 2

**Summary:**

This paper extends and refines existing parameterisation schemes for hyperparameter scaling, most notably CompleteP. The authors introduce/combine several ideas: a continuous-time analysis of SGD to derive scaling rules for the learning rate, weight decay, and other hyperparameters across batch size and token horizon; corrections to inconsistencies in the original completeP work (AdamW $\eps$); per-module and per-depth hyperparameters. They empirically demonstrate that these refinements allow efficient hyperparameter transfer across scaling axes, including model size, batch size, and token count. Impressively, they do this with not only learning rate, but also Adam parameters, initialization scales, and multipliers.

**Strengths:**

Tackling hyperparameter transfer across many scaling axes (width, depth, batch size, and training horizon, etc.) is very ambitious. As far as I know, no prior work has attempted such comprehensive scaling experiments.

The derivation of batch-size and weight-decay scaling from the SDE limit of AdamW is conceptually elegant. The pragmatic “trust-region random search” for finding the per-module multipliers $\alpha_g$ and $\delta_\ell$ seem effective and fairly efficient, showing that the resulting configurations generalize well to large models.

The experiments are extensive and to have combined several ideas regarding hyperparameter scaling make the paper overall very impressive.

**Weaknesses:**

The paper would benefit from a concise quantitative table comparing Complete(d)P vs. CompleteP across representative settings. This would make the incremental gains clearer. Further, it is not totally clear what one ought to compare to, because there is no dedicated related-work section. This makes situating the contribution relative to prior works difficult. There seem to be missing citations, such as [1] (I would also expect to see a quantitative comparison to [1]).

The paper would be more useful to practitioners if it provided recommended base hyperparameters (e.g. base learning rate, initialization scale, etc.) for typical model sizes. Table 1 gives the scaling rules but not concrete base values. In practice, we don't want to run these grid searches (if possible).


[1]: https://openreview.net/pdf?id=DZ6iFdVDrx

**Questions:**

- Can you provide an approximate compute budget (GPU-days) for the per-module hyperparameter search? Section C.1 gives partial details but not the full scope.
- Do you believe these scaling ideas could extend to mid- or post-training adaptation (e.g. LoRA fine-tuning)? Would the same SDE-based scaling rules still hold?
- For practitioners pretraining Transformers from scratch, could you recommend a minimal set of “good defaults” derived from *Complete(d)P*, avoiding the need for exhaustive search?

---

> ### Author Response · Authors · 2025-11-21
> **Rebuttal to Reviewer 36xD 1/1**
>
> Thank you for your time and a thorough review. We are grateful for your encouraging comments on the ambitious nature of our work, and for the quality of our experiments. We welcome this opportunity to address your remaining questions below:
>
> > **The paper would benefit from a concise quantitative table comparing Complete(d)P vs. CompleteP across representative settings.**
>
> We agree quantitatively comparing the proposed major changes to the parameterisation would be valuable. We now compare Complete(d)P to Complete(d)P with width & depth scaling only (i.e. CompleteP with fixes) with a scaling law in the new `Fig. 6`.
>
> We note that it’s important to compare these methods in terms of *scaling laws* (c.f. individual performance values at a fixed scale), as we are comparing rules that determine the *scaling* behaviour. Scaling laws capture how we can expect the performance to change as we scale following a given rule.
>
> We do not compare the width & depth scaling rules of CompleteP to CompletedP as **1)** CompleteP simply didn’t consider QK-norms, so their rules are technically undefined for transformers with QK-norms we consider, and **2)** the remaining modifications are numerically minor. We argue for these remaining modifications on grounds of theoretical correctness.
>
> > **[...] there is no dedicated related-work section.**
>
> We agree it would be beneficial to clarify the connections to related work and prior parameterisations, and have added an “Extended related work” section in the appendix (Appendix E). We now explicitly list what the differences between Complete(d)P vs. prior parameterisations are there.
>
> > **There seem to be missing citations, such as [1](https://openreview.net/pdf?id=DZ6iFdVDrx) (I would also expect to see a quantitative comparison to [1](https://openreview.net/pdf?id=DZ6iFdVDrx)).**
>
> Thanks for pointing [1] out to us! We now discuss it in the related works section (see `L.1225`), as it’s certainly of interest. Their theoretical analysis of why different blocks might need different learning rate multipliers is relevant, and a good motivation for our work.
>
> We believe that their method, however, is not directly comparable to ours. The comparable components of their method are already ablated in our work, and other parts are too costly to directly compare at the scales we consider, as in their work they have to manually tune the hyperparameters at **every** scale. On the other hand, a core focus of our work is *hyperparameter transfer,* and a parameterisation that allows us to scale to much larger models since we don’t have to retune any hyperparameters at large scale.
>
> > **The paper would be more useful to practitioners if it provided recommended base hyperparameters (e.g. base learning rate, initialization scale, etc.) for typical model sizes.**
>
> Our strong recommendation to practitioners would be to tune (per-module) hyperparameters for their specific problem at small scale, and then transfer them to large scale using the proposed rules.
>
> That being said, we are running additional seeds for the hyperparameter search to get a robust estimate for the optimal per-module hyperparameters, and we will add the hyperparameters we identified as important in the appendix for people who want to use them on the dataset we study.
>
> > **Can you provide an approximate compute budget (GPU-days) for the per-module hyperparameter search? Section C.1 gives partial details but not the full scope.**
>
> Thank you for pointing this out, we have now added the compute budget for the search to the appendix in the Appendix D.3.
>
>
> >**For practitioners pretraining Transformers from scratch, could you recommend a minimal set of “good defaults” derived from *Complete(d)P*, avoiding the need for exhaustive search?**
>
>
> As mentioned above, since the hyperparameters could be dataset and setting dependent (e.g. autoregressive vs masked diffusion transformer models), our strong recommendation to practitioners is to retune at small scale, and transfer to larger-scale using the parameterisation we put forward in this work. However, for the autoregressive RedPajama setting we consider in this work, we will provide the hyperparameters we found in the appendix.
>
> > **Do you believe these scaling ideas could extend to mid- or post-training adaptation (e.g. LoRA fine-tuning)? Would the same SDE-based scaling rules still hold?**
>
>
> The SDE-based scaling rules should theoretically still hold in this setting, yes! We think it would be an exciting direction for future work to verify this empirically.
>
> [1] Wang et al., 2025, The Sharpness Disparity Principle in Transformers for Accelerating Language Model Pre-Training

---

> ### Comment · Reviewer_36xD · 2025-11-21
>
> Thank you for your reply!
>
> >We now compare Complete(d)P to Complete(d)P with width & depth scaling only (i.e. CompleteP with fixes) with a scaling law in the new Fig. 6.
>
> Great - Complete(d)P is convincingly better when training on more tokens. To clarify, are you showing CompleteP with the AdamW \eps fix or not?
>
> >and have added an “Extended related work” section in the appendix (Appendix E)
>
> Thank you for including a new related work section.
>
> >Our strong recommendation to practitioners would be to tune (per-module) hyperparameters for their specific problem at small scale, and then transfer them to large scale using the proposed rules.
>
> Can you comment on how small you can go here? Is 50M parameters the limit?
>
> ---
>
> Typos in the new document, section 4:
> - '... could save compute by fasting the best hyperparameters with fewer trials.'
> - 'highglighted'

---

> ### Author Response · Authors · 2025-11-22
> **Responding to Reviewer 36xD's follow-up feedback and questions 1/1**
>
> Thank you for pointing out the typos, and for the clarifying questions! We made the corrections to address the spelling mistakes in the revision.
>
>
> >**Great - Complete(d)P is convincingly better when training on more tokens. To clarify, are you showing CompleteP with the AdamW \eps fix or not?**
>
>
> Indeed, we are showing CompleteP with the $\epsilon$ fix and proper scaling for QK norms (so that it should be a stronger baseline). We have now made that more explicit in the figure caption, thanks!
>
>
> >**Can you comment on how small you can go here? Is 50M parameters the limit?**
>
>
> It's a good question. **We added a new plot to the revision** (new `Fig. 10`) **and a discussion** (new `App. A.1` ) **that sheds some light on how small one can go**.
>
> Going smaller naturally means cheaper HP optimisation, but going too small can break the asymptotic HP transfer principles. We think it's an interesting empirical question at what scale that trade-off becomes suboptimal. `Fig`.` 10` gives insight into where degradations from too small scale in the autoregressive transformer setting on RedPajama. We think it's an exciting avenue for future work to do a thorough characterisation, but, in practice, in transformers, it seems in both ours and prior work (CompleteP) the hyperparameters seem to stabilitise around width ~512 and depth ~12.

---

### Official Review · Reviewer_rHzn · 2025-10-31

**Soundness:** 3
**Presentation:** 2
**Contribution:** 3
**Rating:** 6
**Confidence:** 3

**Summary:**

The paper investigates hyperparameter transfer for pre-training of large Transformer models as model size scales. The proposed approach builds upon μ-parameterization (μP) and its variants, such as Depth-μP, and extends these frameworks to incorporate more recent architectural components like QK-normalization. In addition, the authors broaden their analysis to cover the effects of scaling batch size and training token budget.

The work further examines per-layer hyperparameters, showing that optimizing them independently can yield additional performance improvements. Finally, the paper introduces a local search strategy designed to more effectively navigate the sharp cliffs present in the hyperparameter landscape.

**Strengths:**

- Extending the state of the art (Depth-μP) to include more recent architectural advances such as QK-normalization makes the proposed hyperparameter transfer framework significantly more practical and up to date.

- Addressing hyperparameter transfer across batch size and token budget is both sensible and novel, as this dimension of scaling has received limited attention in prior work.

**Weaknesses:**

- Clarity and notation: Certain sections of the paper are difficult to follow, with inconsistent or undefined symbols. For instance, the variable θ denotes model weights in Section 2 but appears to represent hyperparameters in Section 3. The second paragraph of Section 3.1 in particular is confusing and requires clearer explanations and consistent notation.

- Questionable argument against existing HPO methods: The paper claims that established hyperparameter optimization techniques (e.g., Bayesian optimization) would not meaningfully work in this setting. However, this assertion is not convincingly supported. Even if the loss landscape contains cliffs, Bayesian methods can still locate high-performing regions without fully modeling these discontinuities exactly. This holds particularly for Bayesian optimization methods that do not use Gaussian processes. The authors should provide empirical evidence or an ablation study to substantiate this claim.

- Limited downstream evaluation: The primary metric used is the final validation loss on pre-training data, which the authors argue correlates with downstream performance. While reasonable, showing results on actual downstream tasks (e.g., fine-tuning accuracy or benchmark performance) would strengthen the claim that the transferred hyperparameters provide practical benefits.

**Questions:**

- Line 241: Where does the term \eta * e^k originate from?
- Line 244: What is \Theta_\eta2?

---

> ### Author Response · Authors · 2025-11-21
> **Rebuttal to Reviewer rHzn 1/1**
>
> Many thanks for your detailed review and for the very encouraging comments on identifying the practical impact of our framework. We are grateful for your time reading the paper and this discussion. We have incorporated many of your comments in our revision, and welcome this opportunity to clarify a few aspects of the paper.
>
> > **Clarity and notation: Certain sections of the paper are difficult to follow, with inconsistent or undefined symbols. For instance, the variable θ denotes model weights in Section 2 but appears to represent hyperparameters in Section 3. The second paragraph of Section 3.1 in particular is confusing and requires clearer explanations and consistent notation.**
>
> Thank you for pointing these issues out, we agree that this section was hard to parse.  We changed the symbol for hyperparameters to no longer be theta, but $\zeta$ (zeta) instead. We tried to improve the notation and exposition in Section 3.1 to be more light (see revision, and L.422 in particular), your comments are welcome.
>
>
> >**Questionable argument against existing HPO methods: [..] Even if the loss landscape contains cliffs, Bayesian methods can still locate high-performing regions without fully modeling these discontinuities exactly.**
>
> We agree that these arguments against BO were not needed, and see that they form the main reservation you have on our draft.
>
> Let us clarify that the goal of our paper is not to provide evidence that BO may underperform other HPO methods in the challenging setup (predicting training losses) of this work. Our focus is on the interplay between our completed NN parameterisation and transfer to larger scales: the particular algorithm chosen to carry out HPO is secondary, and we settled on a sturdy approach.
>
> Following your comment, and as stated in our answer to Reviewer **E2dm**, we have toned down these claims *significantly*, rephrasing references to BO (`L.91-95`, `L. 396`, `L. 400`), and welcome your feedback on these updates.
>
> As also mentioned in our response to Reviewer **E2dm**, we did, in fact try to delegate this HP search to a vanilla BO approach, and observed the well-documented [1,2,3] issues that vanilla BO can have in non-stationary, high-dimensional settings.
>
> We also faced a specific difficulty arising from our setup: for many combination of parameters explored by BO, we observed many divergent runs that returned `NaN` loss values. This is wasteful, as it incurs training costs but yields no quantitative knowledge. For instance, we did not show such `NaN` values, e.g., in Figure 8 where `NaN` are replaced by the max loss value, but they do occur often.
>
> >**The primary metric used is the final validation loss on pre-training data, which the authors argue correlates with downstream performance. While reasonable, showing results on actual downstream tasks (e.g., fine-tuning accuracy or benchmark performance) would strengthen the claim that the transferred hyperparameters provide practical benefits.**
>
> We agree that benchmark performance results would enhance the paper, and as in our response to Reviewer **DDMZ**, we are planning to add benchmark performance, notably on the larger models. This will require a few more days but we are hoping to communicate these results before the end of the rebuttal phase.
>
> A disadvantage of benchmark performance is, of course, that they tend to display phase changes (from failure plateau to sudden satisfactory performance), making it much harder to analyse or draw any scaling conclusions.
>
> >**Line 241: Where does the term $\eta * e^k$ originate from?**
>
> $e^k$ were introduced `L.235`. In short, in this illustrative example, $g^k = g + \sigma e^k$ were meant to represent noisy batch gradient, where $g$ is the gradient mean, and $\sigma e^k$ is noise from random mini-batching (modelled as a Gaussian). The term  $\eta e^k$ arises from substituting $g^k = g + \sigma e^k$ in Eq. (1).
>
> >**Line 244: What is $\Theta_{k\eta^2}$?**
>
> $\Theta_t$ is the random variable representing the parameters at time $t$ in the RMSProp SDE. $\Theta_{k \eta^2}$ is then this SDE state at time $t:= k \eta^2$ for some integer $k$. Namely, if we discretise the SDE with step size $\eta^2$, then $\Theta_{k\eta^2}$ for $k=1, 2,...$ are the SDE states at those discretised locations.
>
> [1] Eriksson et al., Scalable Global Optimization via Local Bayesian Optimization, 2019, https://arxiv.org/abs/1910.01739
>
> [2] Snoek et al., Input Warping for Bayesian Optimization of Non-stationary Functions, 2014, https://arxiv.org/abs/1402.0929
>
> [3] Rana et al., High Dimensional Bayesian Optimization with Elastic Gaussian Process, 2017, https://proceedings.mlr.press/v70/rana17a.html

---

### Official Review · Reviewer_DDMZ · 2025-11-01

**Soundness:** 3
**Presentation:** 3
**Contribution:** 3
**Rating:** 6
**Confidence:** 3

**Summary:**

The authors propose Complete(d)P, a refinement of the CompleteP parameterization that extends to modern Transformer components like Query-Key normalization. The key contribution is demonstrating that hyperparameters optimized at a per-module level on small proxy models (50M parameters) can transfer successfully to larger models (1.3B parameters). The paper covers all major optimization hyperparameters, and authors also provide practical guidelines for optimizing the challenging high-dimensional per-module hyperparameter landscape.

**Strengths:**

1) The extension to per-module hyperparameters represents a meaningful advancement beyond prior work on global hyperparameter transfer.

2) Novel weight decay scaling rule for batch size (κ scaling) derived from SDE analysis is theoretically motivated and empirically validated

3) Comprehensive experimental coverage across multiple scaling dimensions (width, depth, batch size, token horizon). Empirical results show a speedup improvement of up to 27%.

**Weaknesses:**

1) Model sizes are relatively limited - while the sizes of the models used in this paper are comparable to CompleteP, µP was evaluated on larger models. Addressing the question of scalability would have strengthened the guildelines proposed by the authors.

2) The evaluation does not include the model's performance on downstream tasks. CompleteP, for example, includes this kind of evaluation. Given that this paper builds directly on said paper, not including this in the evaluation reduces the reader's ability to assess the approach's applicability.

3) Evaluation is conducted only on a Transformer-decoder architecture, so transferability to other Transformer architectures (while certainly possible) is not analyzed.

**Questions:**

1) In Figure 1, what is the "optimal global" baseline? Is this the best global HP found through the random search?

2) You mention that removing QK-norms doesn't completely break transfer (Figure 14), contrary to Dey et al. (2025). What implementation differences might explain this discrepancy?

3) Can you provide error bars or results across multiple random seeds for the main speedup claim (Figure 1)?

4) Feel free to address any of the weaknesses listed above.

---

> ### Author Response · Authors · 2025-11-20
> **Rebuttal to Reviewer DDMZ 1/1**
>
> We are grateful for your reading of the paper, and encouraging comments on the novelty of our per-module HP transfer strategy.
>
> >**Model sizes are relatively limited - while the sizes of the models used in this paper are comparable to CompleteP, µP was evaluated on larger models. [...].**
>
> We agree that scaling up to larger runs would strengthen the message. **We updated Figure 1 with a `1.8B` run trained on up to `26.2B` tokens**, and after making some improvements to HP search at small scale (see general response), we still observe a significant **33% speed-up** compared to the best global HPs at this scale.
>
> As a side-note, these large-scale runs are computationally expensive. μP technically tested on an `6.8B` model, but our analysis is more thorough: they only trained 2 models and compared final losses, whereas we train multiple models to *fit scaling laws* to estimate speed-ups. That said, we agree going even larger would be great: we are running a `7B` comparison, and hope to share the results before the end of the discussion period.
>
> >**The evaluation does not include the model's performance on downstream tasks. CompleteP, for example, includes this kind of evaluation.**
>
> This is a good suggestion. We will try to do a benchmark evaluation, e.g. on MMLU. However, we believe that the main focus of our work must remain on the loss, because
>
> * There is an established precedent and motivation for comparing loss performance in the literature. This consensus stems, e.g., from the Chinchilla [1] paper: *"For simplicity, we perform our analysis on the smoothed training loss which is an unbiased estimate of the test loss, as we are in the infinite data regime (the number of training tokens is less than the number of tokens in the entire corpus)"*
> * There are many works showing that in practice loss is a great near-monotonic proxy for down-stream performance [2,3,4].
>
> That said, we agree that having some down-stream benchmark comparisons would enhance our claims. **We will run standard benchmarks on the largest models when relevant, and do our best to provide this before the rebuttal.**
>
> >**Evaluation is conducted only on a Transformer-decoder architecture, so transferability to other Transformer architectures (while certainly possible) is not analyzed.**
>
> We agree: Investigating the ability of our parameterisation to generalise to other settings would strengthen our empirical analysis and grow readership. However, we chose to prioritize compute on the mainstream setting of AR training to convince our readers of using cutting edge **per-module** approaches.
>
> We are in the process of applying our approach to a Masked Diffusion Model (MDM) with an encoder-only architecture on a different dataset (DCLM). If timing permits, we hope to share the results before the end of the discussion period.
>
> >**In Fig 1, what is the "optimal global" baseline? Is this the best global HP found through the random search?**
>
> Thanks for this comment. We have clarified the caption of Fig. 1, which we agree was lacking in detail.
>
> To rephrase this new caption, we did exactly what you mention. We searched for **6 optimal “global” HPs** at small scale (`50M`, `1.6B` tokens). We budget a total of 3k search runs (details in the appendix). We use a larger budget of runs (5k) for the *per-module* HP search. These results are then transferred to the *same* model size in the middle plot, but transferred to longer training horizon for the global approach. Remarkably, our rightmost plot shows two things:
>
> * Thanks to Complete(d)P, our global HP strategy does transfer successfully to a `600x` FLOPs scale.
> * More crucially, thanks to Complete(d)P, the transfer is *also* successful for the per-module HP strategy, which still outperforms significantly the global strategy.
>
> >**You mention that removing QK-norms doesn't completely break transfer (Figure 14), contrary to Dey et al. (2025). What implementation differences might explain this discrepancy?**
>
> We explored many sources of discrepancy and we are still unsure. None of our changes reproduced the failure of transfer of CompleteP α=0.5. Because Dey et al. (2025) did not release their code (only a reference implementation), we can only guess. We spotted some discrepancies between the parameterisation reported in their paper and their reference implementation (in the AdamW 𝜀 scaling), so it’s possible a bug is to blame.
>
> >**Can you provide error bars or results across multiple random seeds for the main speedup claim (Figure 1)?**
>
> Thanks for this comment. Fig. 1 now includes (on the right) 3 runs per configuration.
>
> [1] Hoffmann et al., “Training Compute-Optimal Large Language Models”, 2022
> [2] Figure 4 in Llama Team, “The Llama 3 Herd of Models”, 2025
> [3] Brandfonbrener et al., “Loss-to-Loss Prediction: Scaling Laws for All Datasets”, 2024
> [4] Chen et al., “Scaling Laws for Predicting Downstream Performance in LLMs”, 2025

---

### Official Review · Reviewer_E2dm · 2025-11-11

**Soundness:** 1
**Presentation:** 3
**Contribution:** 2
**Rating:** 2
**Confidence:** 3

**Summary:**

This work investigates the transfer of optimization hyperparameters of language models from a lower to larger scale. The key contribution is to optimize hyperparameters at the low scale on a per-module-type basis. The per-module optimization, empirically, leads to a speed-up of 27% over global optimization in the large scale training run.

**Strengths:**

* How to set hyperparameters for large models is a crucial question, and investigating per-module hyperparameter optimization on the lower scale is an important idea.
* 27% speedup in the large scale training is a substantial improvement and emprirical evidence of per-module HP transfer being useful is important to a larger audience.
* The most related work is closely discussed throughout the paper and ideas well placed in the literature
* The paper is mostly well written

I did not closely check some of the more theoretical ideas in the paper and may be missing some strengths there.

**Weaknesses:**

* The paper makes a point about Bayesian optimization being an ill fit for the problem setting at many points, but does not include an actual empricial comparison.
* Saying "Our study covers all optimisation hyperparameters of modern models" in the abstract is too bold of a statement. This already breaks down when other optimizers or learning rate schedules are considered, or, e.g., the choice of optimizer becomes a hyperparameter. I suggest making a slight adjustment here.
* Experimental rigor could be improved by considering more seeds and quantifying measurement error, e.g., by showing error bars as the standard error around the mean.
* "Even with this modification, however, we found that this trust-region random search quickly plateaued with a relatively high variance in the final loss values." Showing these results in the paper would be beneficial.
* How much optimization of your search method did you do with respect to your final evaluation metric (validation loss on the 1.3B model). Seeing how your method transfers to scales / models that you have not touched during method development would be good.
* No code is provided to reproduce the experiments. Some details are provided in the Appendix, but I doubt I could reproduce the study.
* The authors do not discuss the limitations of their work.
* Figure 1 "Best grid search loss" is hard to read (even without a red-green blindness). More explanation in the caption would be helpful.
* Figures should make it clear that validation loss is shown
* I would like to see a comparison where only the parameterisation is changed and the search method etc. is kept the same.

**Questions:**

* What do the multiple solid red lines in Figure 1 represent?
* What does "optimal global" refer to, e.g., in Figure 1? Is there a direct comparison to Depth-muP and CompleteP?
* Evaluation is done with respect to final validation loss. Is this also being used to optimize the hyperparameters at the lower scale? If so (maybe additionally) showing results on hold-out dataset would be good.

---

> ### Author Response · Authors · 2025-11-20
> **Rebuttal to Reviewer E2dm 1/2**
>
> Thank you for your thorough review and more generally for your encouraging comments on the relevance of the hyperparameter transfer problem. We are grateful for your time reading our work and our rebuttal. We welcome this opportunity to address your concerns.
>
> > **The paper makes a point about Bayesian optimization being an ill fit for the problem setting at many points, but does not include an actual empricial comparison.**
>
> We understand and agree with your point. In retrospect, we agree that these remarks need to be reworded. They were written in an earlier phase of the project in which we extensively tried *vanilla* Bayesian Optimisation (BO) on our problem to no avail. Specifically:
>
> * The challenges of having vanilla BO work out-of-the-box on the losses we study are well-documented in the literature `[1,2,3]`.
> * We encountered such challenges when trying to get standard BO to work and we ran into numerous and various failures. The problem of selecting hyperparameter search boundaries prevented us from finding good initialisations for Gaussian processes with random search (as is done in standard BO in practice). Our GP fits required a huge amount of samples to look reasonable, each being very costly.
> * We do believe that more advanced BO methods could be leveraged to improve performance. For instance, we believe using trust-region methods, including trust-region variants of BO `[1]`, could be promising. **We highlight this in our new limitation section in `L. 486`.**
>
> Overall, we toned down significantly any claims w.r.t. BO in the revision, as BO is not at the core of our contribution in any way.
>
> `L.91-95`
> *“making random search inefficient and standard Bayesian Optimisation ineffective“ → "These characteristics make it highly inefficient to use random search and have proved very challenging in our experimentations with vanilla Bayesian optimisation“*
>
> *“demonstrate that simple local search strategies” → “opt for simpler local*
> *search strategies“*
>
> `L. 396` [...] *standard Bayesian Optimisation, fail in this regime*. → *struggle in this regime*
>
> `L. 400` Entire sentence changed
>
>
> > **Saying "Our study covers all optimisation hyperparameters of modern models" is too bold of a statement.**
>
> Thanks for this comment. In retrospect, we full-heartedly agree. We settled for this submission title to convey that we were extending transfer to *all* parameters *we* could think of that mattered to practitioners, focusing on those that we felt were overlooked in previous references, such as training duration and batch size.
>
> We agree that it is simpler and wiser to explicitly list these extensions. We changed the title to: ***“Completed Hyperparameter Transfer Across Modules, Width, Depth, Batch & Duration”.*** Subsequently, we also modified the abstract to state: *“Our study covers many of the most important optimisation hyperparameters of modern models...”.*
>
> >**Experimental rigor could be improved by considering more seeds and quantifying measurement error**
>
> We are happy to clarify. Almost all of the experiments we report do use 3 seeds. For some figures, the difference between seeds is so small that they are barely visible and overlap (e.g. if you zoom on `Fig. 2`, width 128, you will notice triplets of points for some LR in the upper left, but less so in other areas). In other Figures, such as `Fig. 8`, they are clearly visible. We have clarified this in all captions. The stability across seeds reflects the well documented stability when training autoregressive models.
>
> The main experiment that did not report more than 1 training seed was `Fig. 1`. We have now updated `Fig. 1` to include 3 seeds.
>
> > **"Even with this modification, however, we found that this trust-region random search quickly plateaued with a relatively high variance in the final loss values." Showing these results in the paper would be beneficial.**
>
> After observing these plateaus, and switching to decaying trust-regions, our implementation has evolved in other aspects.
>
> As a result, to produce a structured ablation that would be comparable to our most recent results (as displayed in the paper), we would need to re-ablate this aspect by reverting to an original constant trust-region setting, which would require significant compute. We feel this would be a waste of resources likely to result mostly in failures. As a result, we hope the reviewer would agree that our qualitative description of the plateauing behaviour suffice to provide the key context and insight to the reader.
>
> >**The authors do not discuss the limitations of their work.**
>
> We have now added a limitations section in the final conclusion section. Please see  revision. We highlight limitations that are informed by this discussion with reviewers.

---

> > ### Author Response · Authors · 2025-11-20
> > **Rebuttal to Reviewer E2dm 2/2**
> >
> > >**How much optimization of your search method did you do with respect to your final evaluation metric (validation loss on the 1.3B model). Seeing how your method transfers to scales / models that you have not touched during method development would be good.**
> >
> >
> > Many thanks for this comment. **This seems like an important misunderstanding of our work**: what you mention as a desirable outcome is exactly what we already show in Figure 1. As a result, we have significantly revised the figure’s caption to make the following clear:
> >
> > **All the per-module hyperparameter search and tuning of the search method were done at the `50M` parameter scale**, training for `1.8B` tokens. Using a search strategy, we obtain a good set of per-module hyperparameters at small scale.
> >
> > **Then**, we start training at larger scale, using the very same hyperparameters. No further tuning was done at larger scale, the hyperparameters were directly transferred from the `50M` scale. The scaled-up results for a `1.8B` model (`1.3B` model in the original submission) in `Fig. 1` are already such a demonstration of transfer to previously unseen scales: we had not trained a single model at this scale before training the models shown in that figure. Furthermore, no large-scale runs (`>50M` scale) were used for the development of the hyperparameter search method either.
> >
> >
> > >**I would like to see a comparison where only the parameterisation is changed and the search method etc. is kept the same.**
> >
> >
> > We agree those would be useful. Our paper makes two contributions: **a)** a complete parameterisation for hyperparameter transfer across different scaling modalities, and **b)** an empirical investigation of per-module hyperparameter transfer. Figure 1 ablates **(b)** by comparing per-module hyperparameters to best global hyperparameters using *the same parameterisation*. To independently ablate the benefits of **(a)**, we included a scaling laws comparison with a fixed set of hyperparameters in the (new) Figure 6. We don’t ablate the effects of the fixes to CompleteP, as they are numerically minor.
> >
> >
> > >**What do the multiple solid red lines in Figure 1 represent?**
> >
> >
> > These are training loss curves from individual training runs to which the scaling laws were fit. We have clarified this in the caption.
> >
> >
> > >**What does "optimal global" refer to, e.g., in Figure 1? Is there a direct comparison to Depth-muP and CompleteP?**
> >
> >
> > “Optimal global” refers to the best “global” (c.f. per-module) hyperparameters that were identified with a thorough hyperparameter search at the 50M parameter scale (same scale as the per-module hyperparameter search). The details are given in Appendix D.2 (previously Appendix C.2). Figure 1 was intended to compare per-module hyperparameters against global hyperparameters, so *both* use the same parameterisation to transfer to larger scale (our Complete(d) parameterisation, including the SDE transfer rules).
> >
> >
> > >**Evaluation is done with respect to final validation loss. Is this also being used to optimize the hyperparameters at the lower scale? If so (maybe additionally) showing results on hold-out dataset would be good.**
> >
> >
> > See the response above: we conduct the extensive hyper parameter search with the validation loss of the *small model* as objective; this search is never informed of the final validation loss of the larger models. We then transfer the hyper parameters that we found to the larger model, and report the validation loss of that model.
> >
> > [1] Eriksson et al., Scalable Global Optimization via Local Bayesian Optimization, 2019, https://arxiv.org/abs/1910.01739
> > [2] Snoek et al., Input Warping for Bayesian Optimization of Non-stationary Functions, 2014, https://arxiv.org/abs/1402.0929
> > [3] Rana et al., High Dimensional Bayesian Optimization with Elastic Gaussian Process, 2017, https://proceedings.mlr.press/v70/rana17a.html

---

### Public Comment · ~Xi_Wang4 · 2025-11-12
**Comments on weight decay v.s. batch size**

Very cool work!

At line 250, the author mentioned that "To the best of our knowledge, we are the first to identify this scaling rule for the weight decay λ", I would like to point out a couple of related recent work that proposes similiar scaling

Appendx F.8 in https://arxiv.org/abs/2411.15958

Also https://arxiv.org/abs/2505.13738

Thanks!

---

> ### Author Response · Authors · 2025-11-20
> **Thanks for sharing these references that we have added and now discuss in our draft.**
>
> Thanks for your interest in our work, and thanks for bringing these relevant papers to our attention. We were not aware that other recent works had explored a similar weight decay rule. We now cite and discuss these works in the paper. More specifically we have changed `L.250` with a statement that more accurately contextualises our contribution:
>
> ***To the best of our knowledge, we are the first to extend the SDE reparametrisation principle [1,2] to the weight decay of AdamW, although we note that the same scaling rule for weight decay was proposed base on other principles in recent work [3,4,5].***
>
> And provide a more detailed discussion in Appendix E that argues why our results have a novel SDE motivation for these principles, especially as compared to Compagnoni et al. [5].
>
> [1] Qianxiao Li, Cheng Tai, and Weinan E. Stochastic modified equations and dynamics of stochastic
> gradient algorithms I: Mathematical foundations.
>
> [2] Sadhika Malladi, Kaifeng Lyu, Abhishek Panigrahi, and Sanjeev Arora. On the sdes and scaling
> rules for adaptive gradient algorithms.
>
> [3] Xi Wang and Laurence Aitchison. How to set adamw’s weight decay as you scale model and dataset size.
>
> [4] Shane Bergsma, Nolan Dey, Gurpreet Gosal, Gavia Gray, Daria Soboleva, and Joel Hestness. Power lines: Scaling laws for weight decay and batch size in llm pre-training.
>
> [5] Enea Monzio Compagnoni, Tianlin Liu, Rustem Islamov, Frank Norbert Proske, Antonio Orvieto, and Aurelien Lucchi. Adaptive methods through the lens of sdes: Theoretical insights on the role of noise

---

> > ### Public Comment · ~Xi_Wang4 · 2025-11-22
> >
> > The updated statement looks very reasonable to me!
> >
> > *Note that [4] also includes some discussion on how \eta * \lambda should be scaled with TPP i.e. training duration, maybe also worth discussion this in your draft.
> >
> > Overall, very nice work! Glad to see more people start to discover the magic of weight decay : )

---

> ### Author Response · Authors · 2025-12-04
>
> Thanks! We're glad you're happy with the way things are phrased in the paper now, and thank you for the kind comments about our work!
>
> > *Note that [4] also includes some discussion on how \eta * \lambda should be scaled with TPP i.e. training duration, maybe also worth discussion this in your draft.
>
> We've also incorporated this point into the discussion of the related work, thanks!
>
> It's worth noting, our paper also effectively covers both learning rate and weight-decay scaling in TPP! We give independent scaling rules for token horizons, and model scale, which means transfer to different TPP ratios holds. We've added a note on this in the related works section.

---

### Author Response · Authors · 2025-12-03

We would like to again thank the reviewers for their feedback and comments.

We have made several additions and changes to the work, which tick many of the boxes the reviewers listed as room for strengthening the paper:
- We added scaling law comparisons to CompleteP, demonstrating our additions and the SDE scaling rules lead to better transfer and better asymptotic loss.
- We clarified our contributions with new title and improved presentation: Complete**d**-P is a novel parameterisation that combines improved Complete-P for transfer across model scale, ***and*** rules for transfer across batch-size and training horizon .
- We clarified the discussion of challenges facing hyperparameter search methods such as Bayesian Optimisation on the per-module hyperparameter loss landscape.
- We added a thorough discussion of related work to better contextualise our changes.
In addition

Crucially, we have now in addition obtained new experimental results that were requested by the reviewers:
- **We have obtained _new 7B training results_ trained for _143B tokens_ (**$13000\times $ **compute cost of the tuning runs) that demonstrate a _>30% speedup_** at this scale.
- We have conducted **new evaluations that show improvements in down-stream performance benchmarks from per-module hyperparameters at 7B scale**: _Arc Easy_ improves from `67.93%` to `68.90%`, _LogicQA 2.0_ improves from `23.79%` to `24.75%`, _HellaSWAG_ from `51.56%` to `52.66%`, _PIQA_ from `73.83%` to `75.73%` and perplexity on _LAMBADA standard_ improves from `565.30` to `528.94`.

**We think these new results and changes significantly strengthen our contribution**, and hope they would address the reviewers' outstanding feedback.

---

### Meta-Review · Area_Chair_WFMT · 2026-01-05

**Summary:**

The authors study hyperparameter transfer across scaling of width, depth, batch size, and training horizon, which is an ambitious goal as some of the reviewers mentioned. Building on CompleteP, the authors added the study of batch size and training horizon using an SDE to model the training dynamics of AdamW, hence allowing the analysis to draw conclusions on batch size and training horizon as well. This work also includes quite large scale experiments, demonstrating that their hyperparameter transfer schemes do indeed work well as they predicts in practice.

Overall, the reviewers complemented the submission for:
- Strong empirical results
- Per module transfer
- Ambitious scope
- SDE based analysis

On the other hand the reviewers found weaknesses in:
- Limitations in experiments (contradicting other reviewers)
- Clarity and presentation
- Lacking careful comparison with prior work
- Lacking open source code for reproducibility

Based on my understanding of the subject, per module transfer was always somewhat known but not carefully studied empirically, although the improvement in training time is impressive. However, the experiments were definitely not small, although one may be able to pick on the exact type of experiments. Either way, as I mentioned before, the authors have clearly demonstrated their scheme does indeed yield hyperparameter transfer, which is a significant contribution on its own. In my opinion, this is the most significant contribution and should alone be sufficient grounds for acceptance.

Due to the unfortunate circumstances of reassigning ICLR ACs, I can no longer interact with the authors, as I have many criticisms and questions remaining. However, I would still like to mention these issues and ask the authors to carefully discuss and update the manuscript in the camera ready, as these are serious concerns, just not enough for me to recommend reject. If I was the AC from the beginning, I would have preferred to ask these earlier on instead.

1. On the SDE Model

I don't fully understand the SDE model of Malladi et al. (2022), and given the central role it plays in this work, I would need further clarifications here. Why is it that $g^k$ is divided by a factor $\sigma$? Wouldn't the size of the update converge to a constant non-zero quantity as the batch size converges to infinity? This model seems to suggest that as batch size $B\to\infty$ the update from the "Adam component" would blow up? This doesn't make a lot of sense to me.

2. On the Batch Size Scaling Rule's Interaction with ABC-Reparameterization

I understand the SDE is not a great model as the authors have mentioned, but it does capture the balance of "scale" between the drift and the diffusion term, or as authors interpret them as signal and noise terms. This does make sense that one would achieve a balance between learning rate and batch size such as the linear scaling in SGD. While I'm not familiar enough with Malladi et al. (2022) on how they achieved a square root scaling, I will take it for granted for now.

However, one thing doesn't make a lot of sense to me, is that the learning rate scale is an artifact of equivalent ABC-reparameterizations. This creates a degree of freedom, where we can allow the learning rate to either increase with width (or depth), or decrease instead, where both regimes would yield the same training dynamics. In these two situations, one would either have an increasing or decreasing batch size to match the learning rate, however the limiting dynamics should be same! This seems like a contradiction to me, and I'm not sure how to make sense of it.

That being said, I also cannot deny the fact that empirically this scaling seems to work, so it is mostly an issue of justification for me.

3. On the Training Horizon Scaling

I don't think the training horizon argument makes sense to me. The reason is that as argued for the batch size scaling previously, the only reason this works is because the time scale between the drift and diffusion term should match. Therefore, the scale of the noise must be roughly speaking $\sqrt{\eta}$ (by choosing batch size) since the drift term is on the order of $\eta$ instead. This naturally puts the time scale of the update at $T = k \eta$ instead of what the authors are suggesting in terms of $T = k \eta^2$, which once again does not make sense to me. As a result, I'm not sure if I believe the argument that is being made here.

It seems to me that the only way this scaling can work is by ignoring the batch size factor (which contradicts the batch scaling), and pretend we ignore the drift term, then the diffusion term has a pre-factor of $\eta$ instead, which naturally puts the time step at $\Delta t = \eta^2$, in order to match the time scale. What this measures is effectively a sense of total amount of noise injected, not exactly a balance based on argument anymore. I hope the authors can explain this to me.

Once again, I'm not arguing against the fact that empirically this scaling works, but the theoretical justification seems lacking in my opinion.

4. On a Principled Approach to Achieve These Scalings

There is a sense that compared to previous work, this work does not identify precise principles such as the desiderata considered by $\mu$P or CompleteP. I believe this is at the core of my theoretical confusion here, since if I understand what is the type of quantities that needs to be preserved during training, this would form a coherent understanding of how hyperparameters should scale across batch size and training horizon.

To summarize all these issues, I hope the authors can use the camera ready (and arXiv) to update the manuscript with these questions and concerns in mind. Given the circumstance is that I am coming in at the very end with a lot of questions, I will not factor in these as part of the decision. And as I mentioned previously, the empirical contributions of verifying the scaling rules for batch size and training horizon alone is sufficient contribution, which is why I will recommend accept.

**Reviewer Concerns:**

Addressed:
- Insufficient Model Scale (Reviewers DDMZ, rHzn)
- Lack of Downstream Evaluation (Reviewers DDMZ, rHzn)

Partially addressed:
- Claims Regarding Bayesian Optimization (Reviewers E2dm, rHzn)
- Mathematical Notation & Clarity (Reviewer rHzn)

Outstanding:
- Statistical Rigor / Error Bars (Reviewer E2dm)
- The "Smallness" Limit of the Proxy Model (Reviewer DDMZ)

**Reviewer Scores:**

Reviewer 36xD: 8
Reviewer DDMZ: 6
Reviewer rHzn: 6
Reviewer E2dm: 2 (potentially raise to 4-5)

---

### Decision · Program_Chairs · 2026-01-26

Accept (Poster)